# The Interplay of Uncertainty Modeling and Deep Active Learning: An Empirical Analysis in Image Classification

**Denis Huseljic**                                                                      *denis.huseljic@uni-kassel.de*
*Intelligent Embedded Systems*
*University of Kassel*

**Marek Herde**                                                                          *marek.herde@uni-kassel.de*
*Intelligent Embedded Systems*
*University of Kassel*

**Yannick Nagel**                                                                        *yannick.nagel@uni-kassel.de*
*Intelligent Embedded Systems*
*University of Kassel*

**Lukas Rauch**                                                                          *lukas.rauch@uni-kassel.de*
*Intelligent Embedded Systems*
*University of Kassel*

**Paulius Strimaitis**                                                      *paulius.strimaitis@student.uni-kassel.de*
*Intelligent Embedded Systems*
*University of Kassel*

**Bernhard Sick**                                                                              *bsick@uni-kassel.de*
*Intelligent Embedded Systems*
*University of Kassel*

**Reviewed on OpenReview:** *https://openreview.net/forum?id=KLBD13bsVl*

## Abstract

Deep *active learning* (AL) seeks to reduce the annotation costs required for training *deep neural networks* (DNNs). Often, deep AL strategies focus on instances where the predictive uncertainty of a DNN is high. Furthermore, Bayesian concepts to model uncertainty are frequently adopted. Despite considerable research, a detailed analysis of the role of uncertainty in deep AL is still missing, especially regarding aleatoric and epistemic uncertainty, both related to predictive uncertainty. This article provides an in-depth empirical study analyzing the interplay of uncertainty and deep AL in image classification. Our study investigates four hypotheses that provide an intuitive understanding of the effects of accurately estimating aleatoric and epistemic uncertainty on existing uncertainty-based AL strategies but also, in the opposite direction, the impact of uncertainty-based AL on the quality of uncertainty estimates that are needed in many applications. By analyzing these hypotheses on synthetic and real-world data, we find that accurate aleatoric estimates can even impair instance selection, while accurate epistemic estimates have negligible effects. Moreover, we provide a publicly available toolbox for deep AL with various models and strategies to facilitate further research and practical applications. Code is available at `https://github.com/dhuseljic/dal-toolbox`.

## 1    Introduction

While deep learning models show exceptional performance in supervised learning tasks such as image classification (He et al., 2016), they require vast amounts of annotated data to achieve high accuracy. The instance annotation process is typically performed by human experts, making it both time-consuming and expensive (Herde et al., 2021). *Active learning* (AL) seeks to minimize this annotation cost by iteratively selecting instances for annotation that are expected to improve the model's performance significantly (Settles, 2009). This enables the model to attain high performance with fewer annotated instances.

Many AL strategies employ an uncertainty-based instance selection and estimate the instances' informativeness using information-theoretic criteria (Lewis, 1995; Gal et al., 2017; Beluch et al., 2018). However, these criteria assume that the model provides accurate predictive uncertainty estimates, e.g., class probabilities. Consequently, standard deep neural networks (DNNs) may not suit that scenario as they do not properly model the two components of predictive uncertainty, i.e., aleatoric and epistemic (Fortuin et al., 2022). While aleatoric (data) uncertainty expresses the inherent noise in the data (Hüllermeier & Waegeman, 2021), e.g., due to noise in the annotation process, epistemic (model) uncertainty reflects the model's limitations in providing trustworthy predictions due to insufficient knowledge (Huseljic et al., 2021). To address this mismatch between DNNs and uncertainty-based AL strategies, research adopts techniques such as *Bayesian neural networks* (BNNs) for AL with the goal to improve the estimation of informativeness and, consequently, AL performance (i.e., accuracy) (Gal et al., 2017; Ren et al., 2022).

Despite considerable research in that field, a detailed analysis of the role of uncertainty remains lacking. Although many studies (Gal et al., 2017; Kirsch et al., 2019) emphasize the importance of epistemic uncertainty in AL, its actual influence is often not thoroughly examined. Specifically, many works (Gal et al., 2017; Kirsch et al., 2019; Pop & Fulop, 2018) claim that epistemic uncertainty plays a critical role in their approaches but neglect to quantify it during AL, for instance, through *out-of-distribution* (OOD) detection metrics. Moreover, the impact of aleatoric uncertainty during AL is frequently overlooked (Pop & Fulop, 2018; Chitta et al., 2018; Ranganathan et al., 2017). As it is a primary component of predictive uncertainty, we would expect it to have a substantial impact in an uncertainty-based selection. Conversely, in specific scenarios, retaining an accurate predictive uncertainty estimation after performing AL may also be necessary, especially in safety-critical domains. However, to the best of our knowledge, this area has only received limited attention so far. Specifically, Beluch et al. (2018) assess the predictive uncertainty through calibration plots at various stages of the AL process, while Pop & Fulop (2018) evaluate calibration after AL. Although both studies provide valuable insights via exemplary studies, a comprehensive numerical analysis will contribute to an in-depth understanding of the topic.

We aim to fill this gap by conducting an extensive empirical study with a focus on image data providing insights into the interplay of uncertainty and deep AL. Our contributions can be summarized as follows:

- We identify and motivate four hypotheses on how aleatoric and epistemic uncertainty influence AL and how an uncertainty-based instance selection impacts their respective uncertainty estimates.

- We conduct *qualitative* studies that investigate AL with (approximately) optimal aleatoric and epistemic uncertainty estimates to analyze the interplay of uncertainty and AL in an ideal setting.

- We present an extensive *quantitative* study of uncertainty-based deep AL strategies using real image data. Specifically, we analyze the effects of aleatoric and epistemic uncertainty on the instance selection and evaluate how these strategies affect uncertainty estimates.

- We provide implementations and propose a toolbox for deep AL with several models and strategies to streamline the research process and help practitioners.

## 2    Background

**Notation**: We consider classification problems in pool-based AL and represent an instance by $\boldsymbol{x} \in \mathcal{X}$ with input space $\mathcal{X}$. In the case of image classification, $\mathcal{X}$ expresses the space of all images. The corresponding

label is represented by $y \in \mathcal{Y}$, where $\mathcal{Y} = \{1, \ldots, K\}$ is the space of $K$ possible labels. A DNN is a function $f_{\boldsymbol{\omega}} : \mathcal{X} \to \mathbb{R}^K$, with network parameters $\boldsymbol{\omega}$, which maps instances to the logit space. Class probabilities for a specific instance can be obtained by applying the softmax function $p(y|\boldsymbol{x}, \boldsymbol{\omega}) = [\text{softmax}(f_{\boldsymbol{\omega}}(\boldsymbol{x}))]_y$. In AL, we assume that we start with a (randomly sampled) initial labeled pool $\mathcal{L} = \{(\boldsymbol{x}_n, y_n)\}_{n=1}^N$ and a large unlabeled pool $\mathcal{U} = \{\boldsymbol{x}_m\}_{m=1}^M$, where $N$ and $M$ denote the number of labeled and unlabeled instances, respectively. We begin the AL cycle by training a randomly initialized DNN on the labeled pool $\mathcal{L}$ using optimization algorithms such as stochastic gradient descent. Subsequently, we estimate informativeness scores for instances in the unlabeled pool $\mathcal{U}$ and query an oracle to annotate the highest-scoring instances. Finally, we add these to the labeled pool with the annotations provided by the oracle and repeat the cycle.

**Aleatoric and Epistemic Uncertainty:** Aleatoric uncertainty arises through *inherent noise* in the data (Hüllermeier & Waegeman, 2021), that is, by collecting labels $y$ from an experiment multiple times with the same instance $\boldsymbol{x}$, we might observe different annotations. We refer to such instances as inherently noisy instances. Meaningful aleatoric uncertainty estimation implies that the probabilistic predictions of a DNN are well-calibrated (Fortuin et al., 2022). Vice versa, we assume that well-calibrated probabilities are also consistent with a good estimation of aleatoric uncertainty. Thus, to improve the estimation of aleatoric uncertainty, we utilize *label smoothing* (LS) (Szegedy et al., 2016; Müller et al., 2019) and *mixup* (Zhang et al., 2018; Thulasidasan et al., 2019) as these techniques do not rely on a calibration dataset and can effectively enhance the calibration of DNNs without incurring additional costs (e.g., annotation cost). We avoid post-hoc calibration methods, such as *temperature scaling*, because these are impractical in AL as they require an additional calibration dataset. Detailed descriptions of label smoothing and mixup are provided in Appendix B.

Epistemic uncertainty arises from insufficient knowledge of the model and can be reduced by incorporating more training data (Hüllermeier & Waegeman, 2021). An accurate estimation of epistemic uncertainty indicates a model's ability to identify instances outside the data distribution (henceforth referred to as OOD) (Liu et al., 2022). Conversely, we hypothesize that a model capable of identifying OOD instances also possesses a strong ability to model epistemic uncertainty (Fortuin et al., 2022). While the probabilistic predictions $p(y|\boldsymbol{x}, \boldsymbol{\omega})$ in case of a DNN include aleatoric uncertainty, they do not model epistemic uncertainty (van Amersfoort et al., 2021; Hüllermeier & Waegeman, 2021; Bengs et al., 2023). The parameters of (deterministic) DNNs are point estimates, precluding the ability to express any uncertainty about them. In contrast, Bayesian models estimate the posterior distributions $p(\boldsymbol{\omega}|\mathcal{L})$ over the parameters of a DNN and incorporate epistemic uncertainty into their predictions by marginalization

$$p(y|\boldsymbol{x}, \mathcal{L}) = \int p(y|\boldsymbol{x}, \boldsymbol{\omega})p(\boldsymbol{\omega}|\mathcal{L})\mathrm{d}\boldsymbol{\omega}. \tag{1}$$

To model epistemic uncertainty, we make use of BNNs, i.e., *Monte-Carlo (MC)-Dropout* (Gal & Ghahramani, 2016), *(deep) ensembles* (Beluch et al., 2018), and *Spectral-normalized Neural Gaussian Processes* (SNGP) (Liu et al., 2020). Simply put, these Bayesian models draw samples from an approximation to the posterior distribution $p(\boldsymbol{\omega}|\mathcal{L})$ and make predictions by estimating the predictive distribution from Eq. 1 via e.g., MC-integration. Detailed descriptions of MC-Dropout, deep ensembles, and SNGP are provided in Appendix B.

**Uncertainty-based Selection Strategies:** In AL, the informativeness of instances is often assessed using a model's output which was trained on the labeled pool $\mathcal{L}$. Here, we focus on selection strategies that utilize uncertainty-based measures as a score for informativeness. A fundamental assumption is that these uncertainty-based measures are beneficial in AL. This implies that probabilistic predictions are required to be better than random guesses by capturing information of the labeled pool $\mathcal{L}$. We distinguish between aleatoric and epistemic strategies based on the model employed (i.e., deterministic or Bayesian). *Aleatoric strategies* are those in which predictive uncertainty is solely based on aleatoric uncertainty. That is, we use predicted probabilities $p(y|\boldsymbol{x}, \boldsymbol{\omega})$ of a deterministic DNN that lacks an epistemic component (Hüllermeier & Waegeman, 2021). In contrast to epistemic strategies (see below), aleatoric strategies are only purely aleatoric due to the combination of informativeness scores with predictions of a deterministic model. We consider the informativeness scores:

- *Least confident* selects instances for which the model outputs a low confidence:

$$\text{LC}[y|\boldsymbol{x},\boldsymbol{\omega}] = 1 - \max_y p(y|\boldsymbol{x},\boldsymbol{\omega}). \tag{2}$$

- *Margin sampling* selects instances for which the model outputs a small difference between the two highest predicted probabilities:

$$\text{MS}[y|\boldsymbol{x},\boldsymbol{\omega}] = 1 - (p(y_1|\boldsymbol{x},\boldsymbol{\omega}) - p(y_2|\boldsymbol{x},\boldsymbol{\omega})), \tag{3}$$

where $y_1$ and $y_2$ are the first and second most probable labels.

- *Model entropy* (Shannon, 1948) selects instances for which the model outputs a high expected uncertainty of probabilistic predictions:

$$H[y|\boldsymbol{x},\boldsymbol{\omega}] = -\sum_{y=1}^{K} p(y|\boldsymbol{x},\boldsymbol{\omega}) \ln p(y|\boldsymbol{x},\boldsymbol{\omega}). \tag{4}$$

*Epistemic strategies* are those in which the predictive uncertainty includes an epistemic component. This is the case when using a Bayesian model, where the predictive distribution takes the form of Eq. 1. We introduce the term epistemic strategies for readability, even though they combine aleatoric and epistemic uncertainty. In those cases, we consider the following informativeness scores commonly used in the literature:

- *Bayesian model entropy* (Gal et al., 2017) is an enhanced version of the predictive entropy incorporating epistemic uncertainty:

$$H[y|\boldsymbol{x},\mathcal{L}] = -\sum_{y=1}^{K} p(y|\boldsymbol{x},\mathcal{L}) \ln p(y|\boldsymbol{x},\mathcal{L}). \tag{5}$$

- *Bayesian active learning by disagreement* (BALD) (Houlsby et al., 2011; Gal et al., 2017) selects instances for which the model outputs high mutual information between label $y$ and its parameters $\boldsymbol{\omega}$. In this context, the mutual information measures the information gained about the parameters $\boldsymbol{\omega}$ by observing a particular label $y$:

$$I[y,\boldsymbol{\omega}|\boldsymbol{x},\mathcal{L}] = H[y|\boldsymbol{x},\mathcal{L}] - E_{p(\boldsymbol{\omega}|\mathcal{L})}[H[y|\boldsymbol{x},\boldsymbol{\omega}]]. \tag{6}$$

To estimate the expectation, MC integration is employed here (Gal et al., 2017).

- *Variation ratio* (Beluch et al., 2018) selects instances for which a model outputs a diverse set of class predictions:

$$\text{VR}[\boldsymbol{x}] = f_m/T, \tag{7}$$

where $f_m$ is the number predictions that are not the modal class and $T$ is the total number of sampled predictions, such as the number of ensemble members.

Note that we do not examine uncertainty-based selection strategies that account for instances' diversity in batch selection, such as BatchBALD (Kirsch et al., 2019) or BADGE (Ash et al., 2020). We leave an in-depth investigation of those techniques for future work.

**Metrics:** Recall that we assume that the calibration of a model, i.e., the calibration of its predictive distribution, is typically used to indicate the quality of aleatoric uncertainty estimates. Accordingly, we evaluate the calibration of DNNs with commonly used metrics (Ovadia et al., 2019; Kumar et al., 2019; Nixon et al., 2019) to obtain numerical estimates. On the one hand, we use the proper scoring rules *negative log-likelihood* (NLL) and *Brier score* (Brier), as they induce a calibration measure (Ovadia et al., 2019). On the other hand, we also measure *top-label* and *adaptive calibration errors* (TCE and ACE), as these are more intuitive to interpret (Kumar et al., 2019; Nixon et al., 2019).

To assess epistemic uncertainty estimates, we adhere to common techniques from the literature (Fortuin et al., 2022) and rely on metrics from OOD detection. In this context, we assume a binary classification problem where we distinguish between *in-distribution* (ID) and OOD instances. In particular, we consider the *area under the precision-recall curve* (AUPR) to evaluate a model's ability to detect OOD instances (Liu et al., 2022). Consequently, we require a separate OOD dataset in addition to the test dataset. Depending on the model, we calculate the appropriate entropy measure from Eq. 4 or 5 for each instance of both datasets. Subsequently, we employ the AUPR to evaluate whether instances' entropies can be used to distinguish between ID and OOD. Ideally, models with good epistemic estimates should produce high-entropy predictions for all OOD instances.

In addition, we assess the generalization of models by employing the accuracy. A detailed definition of all calibration, OOD detection, and generalization metrics can be found in Appendix A.

## 3 Hypotheses

We now define four hypotheses addressing the influence of aleatoric and epistemic uncertainty on AL and the effect of AL on these uncertainties. Additionally, for each hypothesis, we provide the underlying motivations and intuitions. The first two hypotheses investigate aleatoric uncertainty only, excluding the influence of epistemic uncertainty. Thus, we focus on deterministic models aiming at modeling aleatoric uncertainty. The remaining two hypotheses consider Bayesian models and, thus, mainly focus on epistemic strategies. They address the influence of incorporating an epistemic component into the predictive distribution.

**Influence of Aleatoric Uncertainty on Instance Selection:** One would expect that an accurate estimation of aleatoric uncertainty is an essential factor in AL, especially because the predicted probabilities of a model are employed to determine the instances' informativeness scores. However, we argue that an accurate estimation does not necessarily improve instance selection when considering selection strategies that purely focus on aleatoric uncertainty. For example, a selection based on aleatoric uncertainty, such as the one using Eq. 4, has the property to only focus on difficult regions of the input space near the decision boundary. Improving aleatoric estimates only induces a change in instance selection when there is a change in the ordering of the informativeness scores of instances. Typically, this is not the case with most calibration methods, as these only lead to minor changes in predicted probabilities. However, when calibration methods strongly impact the predicted probabilities, the selection will place a stronger emphasis on inherently noisy instances near the decision boundary. This results in a more challenging learning problem. Hence, good aleatoric uncertainty estimates may even aggravate the learning problem, which can lead to sub-optimal optimization of the model. In such cases, having a large initial pool is vital to ensure that only a refinement of the decision boundary is needed. Vice versa, inaccurate estimation of aleatoric uncertainty may lead to a more diverse selection. For example, an imprecise estimation might cause a selection of instances that are not inherently noisy, thereby including instances helpful for the learning problem. As we assume that a well-calibrated model reflects a good estimation of aleatoric uncertainty, we hypothesize that for AL strategies that only consider aleatoric uncertainty in their selection, improving calibration will not lead to an improved instance selection and can even potentially hurt AL performance. Therefore, we formulate the following hypothesis:

> **Hypothesis 1 (H1):** A good estimation of aleatoric uncertainty does not improve the instance selection of aleatoric strategies. Instead, it may even deteriorate the selection by biasing it towards inherently noisy instances that are difficult to learn.

**Influence of Aleatoric Strategies on Aleatoric Uncertainty Estimates:** A common assumption when training DNNs is that data is independent and identically distributed (i.i.d.). While this may be true when randomly acquiring a dataset, this assumption is violated when selecting instances through an AL strategy. Therefore, we hypothesize that uncertainty-based active instance selection influences the aleatoric uncertainty estimates of a model. We argue that an AL selection focusing solely on aleatoric uncertainty can improve the respective model estimates. Specifically, DNNs are known to output overconfident probabilistic predictions when training with instances that are i.i.d. (Guo et al., 2017). As the selection focuses on difficult regions with inherently noisy instances, this mimics regularization and, consequently, counteracts the overconfidence

of DNNs. When model complexity is low – either due to the model's limitations or increased difficulty of the task – this form of regularization may be less beneficial. This motivation follows the theoretical work from Farquhar et al. (2021). By focusing on difficult instances, the network becomes more uncertain, leading to a better estimation of "true" uncertainties. From this motivation we derive the following hypothesis:

> **Hypothesis 2 (H2):** Aleatoric selection strategies improve a model's aleatoric uncertainty estimates by leveraging inherently noisy instances to counteract DNNs' overconfidence.

**Influence of Incorporating Epistemic Uncertainty into AL:** Epistemic uncertainty reflects the knowledge of a model, and thus, it has already been argued that this kind of uncertainty is essential for AL. Intuitively, querying instances from the input space where the model has limited knowledge may reduce its own epistemic uncertainty more efficiently. In this regard, incorporating epistemic uncertainty into an uncertainty-based selection strategy can be seen as introducing an explorative component. This is because selecting the most uncertain instances is akin to focusing on OOD instances, which emphasizes the exploration of regions in the input space that were not seen during training. Consequently, we hypothesize that this is also the reason why the incorporation of epistemic uncertainty should improve AL performance. It is important to mention that drawing conclusions based on epistemic uncertainty alone is challenging. Approaches that model epistemic uncertainty, such as BNNs, enhance the predictive uncertainty. This may also affect the aleatoric part of the predictive distribution. Accordingly, improvements in AL might not be attributed solely to the epistemic component but could also be due to the combination of aleatoric and epistemic uncertainty. From this motivation, we derive the following hypothesis:

> **Hypothesis 3 (H3):** The additional incorporation of epistemic uncertainty into AL improves performance by introducing an explorative component into the selection, which helps to recognize regions that were never seen during training.

**Influence of Epistemic Strategies on the Predictive Uncertainty:** Additionally incorporating epistemic uncertainty encourages a selection of instances in regions about which the model has limited knowledge. Consequently, this selection should facilitate quick exploration of potential model parameters capable of effectively learning the task. Accordingly, we assume that uncertainty-based AL selections considering epistemic uncertainty improve the respective estimation, and thus predictive uncertainty. In fact, the primary goal of many proposed AL strategies (Gal et al., 2017; Beluch et al., 2018; Kirsch et al., 2019) is to reduce epistemic uncertainty efficiently. To validate this, we analyze the following hypothesis:

> **Hypothesis 4 (H4):** Epistemic selection strategies improve the model's aleatoric and epistemic estimates, as they promote selecting instances in regions where the model has limited knowledge.

We want to emphasize that a hypothesis might correlate with another one if true. In particular, this means that the effect between hypotheses can be contrary but also of varying severity. In the following experiments, we aim to investigate these hypotheses independently.

## 4 Experiments

We analyze the proposed hypotheses using both synthetic and real-world datasets for a qualitative and quantitative evaluation. For more detailed information on implementations, we refer to Appendix D.

### 4.1 Synthetic Data

We start with a simple illustrative 2D example with a linear model to gain intuitions about the impact of aleatoric and epistemic uncertainty on AL. Following this, we investigate if these intuitions can also be transferred to image classification by introducing a synthetic image dataset. In this context, our analysis

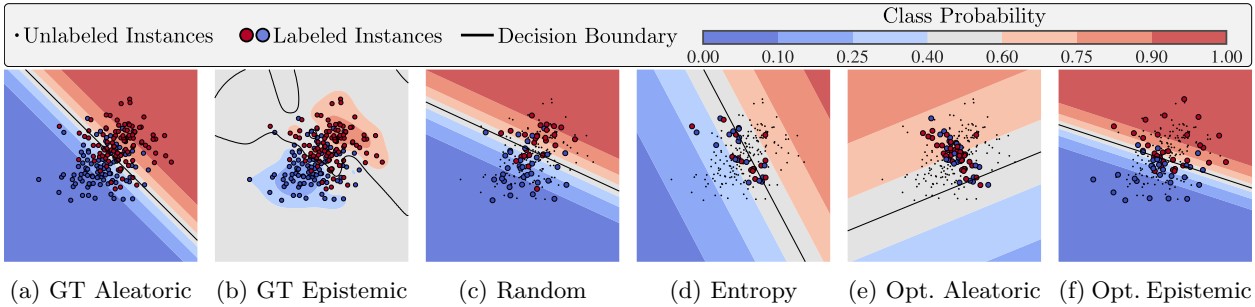

(a) GT Aleatoric    (b) GT Epistemic    (c) Random    (d) Entropy    (e) Opt. Aleatoric    (f) Opt. Epistemic

Figure 1: AL experiment illustrating the behavior of entropy sampling with optimal estimates for aleatoric and epistemic uncertainty. (a) GT aleatoric uncertainty from data generating distribution $p(\boldsymbol{x}, y)$. (b) GT epistemic uncertainty approximated by a GP. (c) Linear model trained on randomly selected instances. (d) Linear model trained on high-entropy instances using the model's predicted probabilities. (e) Linear model trained on high-entropy instances using GT aleatoric probabilities. (f) Linear model trained on high-entropy instances using GT epistemic probabilities.

focuses on convolutional neural networks, which are prevalent for computer vision tasks. Statements related to one of the four hypotheses are indicated by (H1) – (H4) to facilitate readability.

**Illustrative Example:** We consider a 2D dataset consisting of two Gaussian distributions with a strong overlap of the probability masses indicating high aleatoric uncertainty. A crucial advantage of this setup is the precise determination of aleatoric uncertainty, as the true class probabilities $p(y|\boldsymbol{x})$ are accessible. This is due to the ability of controlling the data generating distribution $p(\boldsymbol{x}, y) = p(y|\boldsymbol{x})p(\boldsymbol{x})$. Since epistemic uncertainty is model-dependent and lacks a *ground truth* (GT), we approximate it using a Gaussian process as commonly done in the literature (Liu et al., 2022). Consequently, we use the knowledge of GT aleatoric and (approximate) GT epistemic uncertainty to perform active instance selection. Further details regarding the dataset and AL settings can be found in Appendix E. Figures 1 (a) and (b) display the dataset with corresponding GT class probabilities $p(y|\boldsymbol{x})$ and $p(y|\boldsymbol{x}, \mathcal{L})$ represented as contours. In the case of GT aleatoric uncertainty, overlapping regions exhibit high aleatoric uncertainty (gray color), as class probabilities tend to be close to uniform. In contrast, high GT epistemic uncertainty arises not only along the decision boundary, but also in regions where instances are scarce or unavailable. Subsequent plots illustrate a linear model's decision boundary after AL, trained on the highlighted labeled instances. Random sampling, shown in Fig. 1 (c), selects instances with equal probability, while model entropy, shown in Fig. 1 (d), selects informative instances based on Eq. 4 using the linear model's predicted probabilities. Opt. aleatoric and opt. epistemic entropy, shown in Fig. 1 (e) and (f), employ model entropy (Eq. 4) and Bayesian model entropy (Eq. 5) but leverage the optimal probabilities $p(y|\boldsymbol{x})$ and $p(y|\boldsymbol{x}, \mathcal{L})$.

Notably, random sampling results in a model closely resembling the GT (cf. GT aleatoric). In contrast, model entropy sampling leads to a decision boundary that roughly aligns with the GT, but seems to estimate predictive probabilities with higher predictive uncertainty (cf. hypothesis H2). This is illustrated by the contours which indicate that the model's predicted probabilities cover a substantial region of the input space with values between 0.25 and 0.75. The first surprising observation is that the model arising from opt. aleatoric entropy completely fails to learn effectively which is likely due to the strong inherent noise of selected instances with a GT probability of 0.5 (H1). Finally, opt. epistemic entropy yields a well-performing model with instances rather evenly dispersed throughout the dataset, emphasizing exploration (H3). Furthermore, the predicted probabilities exhibit the least amount of uncertainty, which might be ascribed to the primary objective of rapidly reducing the epistemic uncertainty (H4).

***Takeaway:*** *This simple example emphasizes how focusing on highly noisy instances through an optimal estimation of aleatoric uncertainty may lead to unsatisfactory models. Moreover, it demonstrates that incorporating different types of uncertainty with different qualities of their estimates can lead to distinct predictive distributions, as shown by model entropy.*

**Image Data:** We now consider an experiment involving artificial image data to analyze our hypotheses on a more complex model, i.e., ResNet18 (He et al., 2016). Similar to the 2D case, we focus on a binary classification problem and construct a synthetic dataset with access to the ground truth class probabilities $p(y|\boldsymbol{x})$. Specifically, for each RGB image and each channel in that image, we set a random amount of pixels at random positions to their maximum values. The corresponding labels are then drawn from a Bernoulli distribution with the probability dependent on the relative sum of all pixel values. This data generating procedure reflects inherent noise in data, that is, aleatoric uncertainty. The ground truth mapping of the relative pixel sum to its probability $p(y|\boldsymbol{x})$ is shown in Fig. 2 (a). The highest attainable accuracy is 75%. Intuitively, the task involves distinguishing "black" from "white" images, with different "RGB noise" levels specifying the amount of aleatoric uncertainty. An instance with low aleatoric uncertainty is depicted in Fig. 2 (b). It has a high relative pixel sum indicating a high GT probability $p(y = 1|\boldsymbol{x})$. In contrast, an instance exhibiting high aleatoric uncertainty, shown in Fig. 2 (c), corresponds to an image where nearly 50% of its pixels are set to one. Even though this synthetic dataset predominantly consists of random RGB noise, access to the GT probabilities enables us to analyze AL with true aleatoric uncertainty estimates within the image domain. As it is infeasible to obtain GT epistemic uncertainty, we once again employ an approximation. In particular, we use deep ensembles as they have shown to provide accurate estimates in the context of image classification (Ovadia et al., 2019; Liu et al., 2022).

We start AL by randomly initializing the labeled pool $\mathcal{L}$ with one instance per class. Next, we train a randomly initialized ResNet18 on the labeled pool and select two instances for annotation in each cycle. Analogous to the 2D example, we employ random sampling, model entropy as per Eq. 4, and its optimal counterparts based on GT aleatoric and (approximate) GT epistemic uncertainties. We repeat this process until a budget of 100 instances is reached. Figure 3 reports accuracy, NLL, and Brier score on an independent test split for each cycle. As we have access to the GT probabilities, we replace calibration errors with Brier scores, which do not require binning.

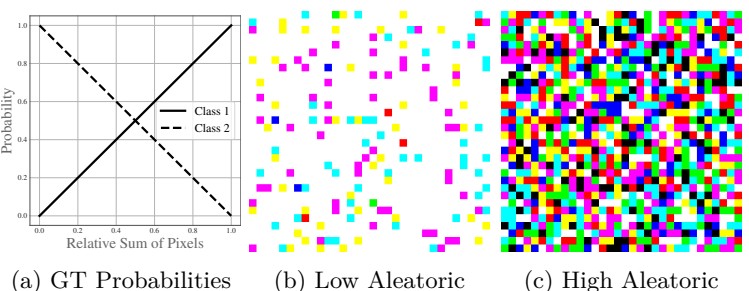

(a) GT Probabilities    (b) Low Aleatoric    (c) High Aleatoric

Figure 2: Synthetic image dataset with ground truth class probability $p(y|\boldsymbol{x})$ based on the relative pixel sum. (a) Mapping of relative pixel sum to probabilities $p(y|\boldsymbol{x})$. (b) Low aleatoric instance with a high relative pixel sum. (c) High aleatoric instance with about 0.5 relative pixel sum.

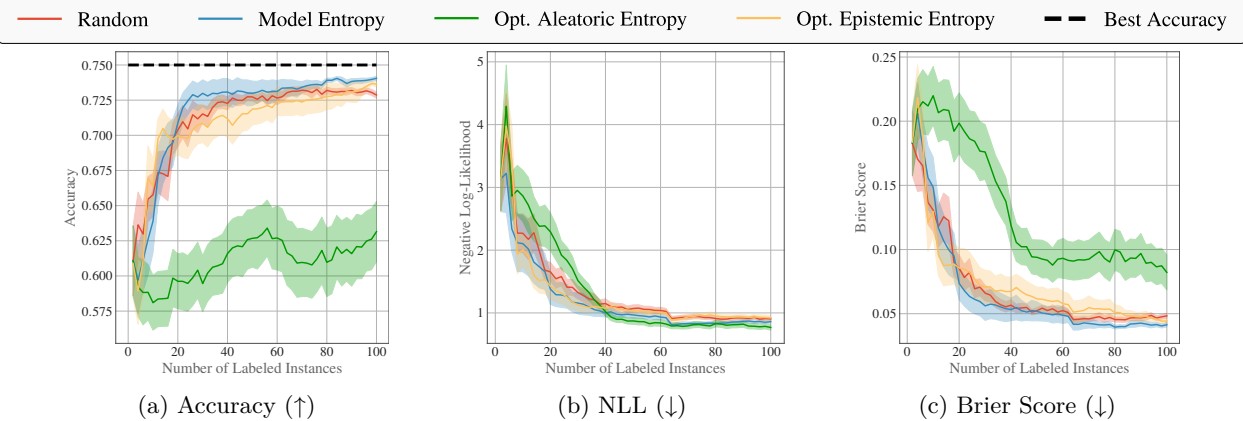

(a) Accuracy (↑)      (b) NLL (↓)      (c) Brier Score (↓)

Figure 3: Learning curves for the synthetic image dataset, showing (a) accuracy, (b) negative log-likelihood, and (c) Brier score. Results are averaged over 25 AL repetitions (standard errors as shaded regions).

Examining the accuracy in Fig. 3 (a), we see that random, model entropy, and opt. epistemic entropy select instances that enable the model to learn well. Notably, model entropy attains a higher accuracy

than random and opt. epistemic entropy, which is potentially due its goal of refining the decision boundary. The explorative component of random and opt. epistemic entropy might contribute to this small accuracy difference. Additionally, opt. epistemic uncertainty outperforms random towards the end, likely because it focuses on the decision boundary once the epistemic uncertainty is sufficiently decreased. Interestingly, both random and opt. epistemic entropy slightly outperform model entropy initially, potentially because exploration is a critical factor in the beginning of AL. In contrast, the opt. aleatoric entropy selection struggles to learn in this synthetic example (cf. low accuracy values). Similar to the 2D case, this issue is likely due to the strong inherent noise of the instance's labels making it difficult to learn (H1). When assessing the NLL in Fig. 3 (b), all strategies seem to improve their predictive estimates throughout AL. Despite struggling to attain competitive accuracy, opt. aleatoric entropy surprisingly yields the most accurate estimates, possibly due to underfitting in conjunction with the NLL. Furthermore, model entropy also performs well, indicating that an implicit regularization through noisy instances can enhance predictive uncertainties. Regarding the Brier score in Fig. 3 (c), model entropy offers the best-calibrated model (H2). Aligning with H2, selecting noisy instances seems to benefit calibration. However, focusing on highly noisy instances may also deteriorate calibration, as shown by opt. aleatoric entropy. While the calibration of random and opt. epistemic entropy are only slightly worse, incorporating epistemic uncertainty seems to be negligible within this problem.

***Takeaway:*** *We conclude that, within the context of this binary image classification problem, model entropy based sampling demonstrates superior performance in terms of NLL, Brier score, and accuracy. Although the instance selection of opt. epistemic entropy also yields well-working models, the incorporation of epistemic uncertainty seems negligible. Furthermore, this experiment suggests that an enhanced estimation of aleatoric uncertainty may not be advantageous for aleatoric strategies.*

## 4.2 Real-World Data

We conduct several experiments on real image classification datasets to further analyze the previously stated hypotheses. For all experiments, we adhere to recent recommendations from the literature for evaluating AL performance (Munjal et al., 2022; Hacohen et al., 2022). Unless stated otherwise, we use a ResNet18 architecture again.

### 4.2.1 Setup

**Datasets:** We analyze our hypotheses on multiple datasets that are commonly used in the deep AL literature (Munjal et al., 2022; Hacohen et al., 2022). However, to provide a clear and extensive analysis of each hypothesis, we primarily focus on CIFAR10 (Krizhevsky, 2009) here. The remaining experiments can be found in Appendix C. To calculate metrics assessing epistemic uncertainty (i.e., AUPR), we require OOD datasets for each respective ID dataset. For CIFAR10, we employ CIFAR100 (Krizhevsky, 2009) and SVHN (Netzer et al., 2011) as OOD datasets to evaluate the model's near- and far-OOD detection capabilities (Ren et al., 2021), respectively. This approach ensures a comprehensive assessment of epistemic uncertainty estimates. Detailed descriptions of each dataset, along with their respective preprocessing steps, are provided in Appendix F.

**AL Settings:** In every cycle, we train a DNN for 200 epochs with a batch size of 32. For CIFAR10, we start with a randomly initialized labeled pool of 128 instances and acquire 128 instances per cycle. After 38 AL cycles, we reach a budget of 4992 instances. To assess the behavior of models during AL, we report *area under the learning curves* (AUC) for each metric. Additionally, we provide final metrics after AL. All learning curves are averaged over ten repetitions.

**Hyperparameters:** Recently, Munjal et al. (2022) emphasized the significance of hyperparameter selection for deep AL. Their work demonstrated that the choice of learning rate and regularization significantly influence AL performance. We observed similar behavior in our experiments. Especially the hyperparameters of techniques aiming to improve the predictive uncertainty (e.g., the amount of label smoothing) required careful tuning in conjunction with learning rate and weight decay. As our objective is to examine the behavior of uncertainty and deep AL, we aim at establishing an optimal setting by utilizing the best combination of hyperparameters for a specific budget. To this end, we consider a randomly sampled training dataset corresponding to the respective total budget and a separate validation dataset to optimize hyperparameters.

Table 1: Quality of aleatoric uncertainty estimates of selection models on CIFAR10 during AL in form of *AUC*.

| | NLL ($\downarrow$) | Brier ($\downarrow$) | TCE ($\downarrow$) | ACE ($\downarrow$) |
|---|---|---|---|---|
| | Random | | | |
| - | 1.070 | 0.425 | 0.138 | 0.074 |
| LS | 0.926 | 0.395 | 0.073 | 0.032 |
| Mixup | 0.824 | 0.358 | 0.049 | 0.028 |
| | Least Confidence | | | |
| - | 1.018 | 0.407 | 0.123 | 0.066 |
| LS | 0.883 | 0.380 | 0.056 | 0.028 |
| Mixup | 0.810 | 0.355 | 0.056 | 0.027 |
| | Margin | | | |
| - | 0.998 | 0.398 | 0.124 | 0.068 |
| LS | 0.864 | 0.370 | 0.059 | 0.028 |
| Mixup | 0.786 | 0.342 | 0.050 | 0.025 |
| | Entropy | | | |
| - | 1.038 | 0.415 | 0.126 | 0.067 |
| LS | 0.889 | 0.384 | 0.058 | 0.029 |
| Mixup | 0.813 | 0.356 | 0.057 | 0.028 |

Table 2: *Final / AUC* accuracies ($\uparrow$) of evaluation models on CIFAR10 for different aleatoric strategies and calibration methods.

| | Random | Least Confident | Margin | Entropy |
|---|---|---|---|---|
| - | 0.849 / 0.720 | **0.872** / 0.731 | **0.873** / 0.738 | **0.872** / 0.726 |
| LS | 0.849 / 0.719 | 0.871 / 0.732 | 0.873 / 0.737 | 0.869 / 0.729 |
| Mixup | 0.851 / 0.719 | 0.867 / 0.717 | 0.869 / 0.728 | 0.866 / 0.717 |

Table 3: *Final* aleatoric estimates (Brier $\downarrow$ / TCE $\downarrow$ / ACE $\downarrow$) of evaluation models on CIFAR10 for different aleatoric strategies and calibration methods.

| | - | Label Smoothing | Mixup |
|---|---|---|---|
| Random | 0.231 / 0.065 / 0.052 | 0.231 / 0.065 / 0.052 | 0.230 / 0.064 / 0.052 |
| Least C. | 0.197 / 0.048 / 0.044 | 0.200 / 0.050 / 0.045 | 0.206 / 0.055 / 0.049 |
| Margin | 0.196 / 0.052 / 0.046 | 0.194 / 0.051 / 0.046 | 0.202 / 0.055 / 0.048 |
| Entropy | 0.197 / 0.050 / 0.045 | 0.202 / 0.051 / 0.046 | 0.208 / 0.057 / 0.051 |

Note that this way of hyperparameter optimization is not possible when employing AL in the real world, as neither the training nor validation datasets would be available. Hyperparameter optimization in the field of deep AL remains an open problem and harms its practicability. We provide more details on the optimization and model training procedure in Appendix F.

**Model Comparison:** In an AL setting where models employ techniques to improve uncertainty estimation, comparing AL selection strategies becomes challenging. For example, mixup's data augmentation technique improves not only predictive uncertainty estimation, but also accuracy. To disentangle potential improvements resulting from uncertainty modeling techniques (aleatoric or epistemic) and from active uncertainty-based instance selection, we utilize two models for evaluating AL. We will have a model solely responsible for selecting instances while it is enhanced with techniques that improve uncertainty estimation. We refer to this model as *selection model*. It is typically used in the literature and its performance serves as a basis for comparing different selection strategies (Gal et al., 2017; Beluch et al., 2018). However, this comparison will not reveal whether any improvements are due to the instance selection or the applied uncertainty modeling techniques. For this purpose, we introduce an additional model responsible for evaluation. We refer to this model as *evaluation model*. It is trained on the selected instances of the selection model but does not use any techniques to improve its uncertainty estimates. Using an evaluation model allows us to draw conclusions while ensuring that mainly the instance selection leads to any differences. We refer to Sec. 5 for further discussions on that topic.

### 4.2.2 Results

We now present the findings of our experiments and evaluate the hypotheses outlined in Sec. 3 by delving into a comprehensive comparison of all strategies, presented in a tabular format. Although we only report the results of CIFAR10 here, the behavior of the models is consistent across several investigated datasets, consolidating the inferred takeaways. Moreover, to facilitate the discussions below, we only report key results from selection and evaluation models. Further tables on all datasets can be found in Appendix C.

**H1:** To evaluate H1, we first confirm that the calibration methods enhance aleatoric estimates and subsequently assess the evaluation model's accuracies. Using accurate estimates for selection should enhance the accuracy of the evaluation models. Table 1 presents the quality of aleatoric uncertainty estimates during AL of the selection model. Based on the AUC values of NLL, Brier score, TCE, and ACE of the selection models, we observe that predictive uncertainty and calibration of selection models are consistently enhanced by calibration methods during AL. This implies that the instance selection is based on improved aleatoric uncertainty estimates. An example is shown in Fig. 4 (b), where we observe improved calibration compared to the non-calibrated model (blue) throughout AL. Following H1, we anticipate that a more accurate estima-

tion does not improve but might even lead to a worse instance selection that can result in subpar evaluation model results. Inspecting the final and AUC accuracies in Tab. 2, we find that this indeed occurs. There are no noticeable accuracy improvements of evaluation models using the selection of label smoothing, even though the informativeness scores are more accurate. Furthermore, investigating the selection of mixup, evaluation models even perform worse compared to utilizing the non-calibrated selection. For example, entropy sampling of mixup exhibits lower evaluation accuracy than entropy sampling of the non-calibrated model, despite the enhanced aleatoric uncertainty estimates. Figure 4 presents exemplary learning curves that depict accuracies of evaluation models and calibration errors of selection models with the random and margin sampling strategies. Although the selection ACE indicates enhanced calibration compared to the non-calibrated model throughout AL, we cannot observe any noticeable benefits in evaluation accuracy by improving calibration. In fact, margin sampling using the best-calibrated model for selection appears to perform the worst in terms of evaluation accuracy.

***Takeaway: H1 is valid.*** *Improving aleatoric uncertainty estimates using dedicated calibration methods does not enhance instance selection in deep AL. Instead, it might lead to a worse instance selection.*

**H2:** Table 3 shows that each aleatoric strategy leads to an enhanced Brier score, TCE, and ACE compared to random. Further analyses in Appendix C based on the overconfidence error (Thulasidasan et al., 2019) reveal that the resulting models are less overconfident. These findings corroborate the theoretical study from Farquhar et al. (2021), which suggest that AL is a form of regularization. This suggests that these strategies improve predictive uncertainty and calibration compared to random sampling, aligning with our hypothesis. Moreover, this improvement is most prominent in the case of the non-calibrated model, where no calibration technique is employed. In contrast, we observe a lower improvement over random sampling when comparing mixup's aleatoric estimates. We believe this is due to an excessive focus on highly aleatoric instances. Additionally, it may also be caused by differences in training. Figure 4 (c) depicts exemplary learning curves of the ACE, showing that aleatoric strategies lead to an improved calibration.

***Takeaway: H2 is valid.*** *Uncertainty-based AL strategies improve predictive uncertainty and calibration of DNNs by incorporating inherently noisy instances that counteract overconfidence of models.*

**H3:** We expect that incorporating (high quality) epistemic uncertainty into AL improves the selection of instances and, therefore, evaluation accuracy. However, comparing the evaluation accuracies of the non-epistemic model in Tab. 2 with the epistemic strategies in Tab. 5, we observe no meaningful improvement, suggesting epistemic uncertainty hardly impacts AL selection. Even by directly comparing evaluation accuracies of entropy selections with and without epistemic uncertainty, we see inconsistencies. Table 4 presents the quality of aleatoric and epistemic uncertainty estimates of Bayesian selection models during AL in form of AUC values. We see that ensembles consistently outperform MC-Dropout and SNGP in terms of Brier score, ACE, and CIFAR100 AUPR, indicating superior predictive and epistemic uncertainty estimates. In contrast, MC-Dropout yields the worst epistemic uncertainty estimates of the considered models. Consequently, following H3, we expect that the instance selection of ensembles will be better than that of MC-Dropout and

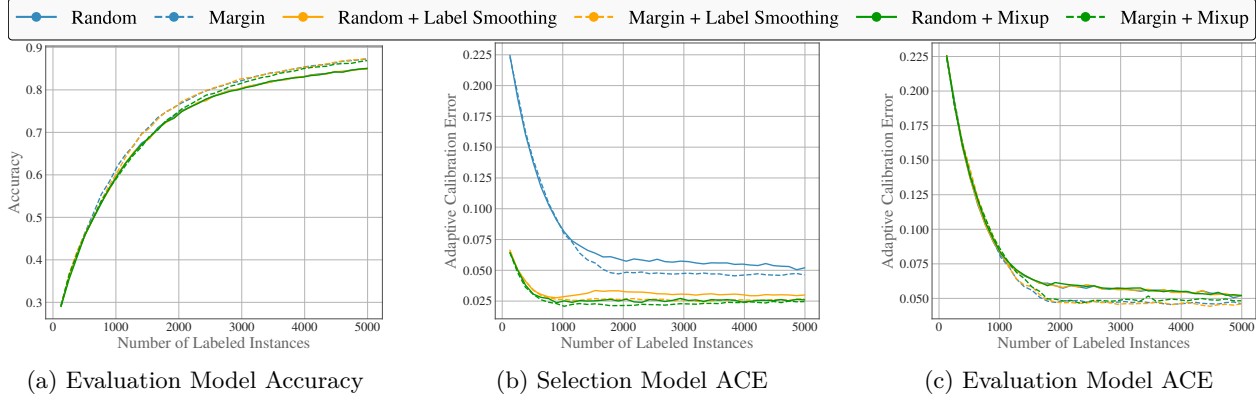

(a) Evaluation Model Accuracy      (b) Selection Model ACE      (c) Evaluation Model ACE

Figure 4: Learning curves on CIFAR10, averaged over 10 seeds.

Table 4: Quality of aleatoric and epistemic estimates of Bayesian selection models on CIFAR10 during AL in form of *AUC*.

|  | Brier (↓) | ACE (↓) | AUPR (↑) CIFAR100 | AUPR (↑) SVHN |
|---|---|---|---|---|
| Random | | | | |
| MC-Dr. | 0.416 | 0.049 | 0.688 | 0.803 |
| SNGP | 0.425 | 0.038 | 0.703 | 0.837 |
| Ensemble | 0.374 | 0.046 | 0.706 | 0.804 |
| Bayesian Entropy | | | | |
| MC-Dr. | 0.416 | 0.052 | 0.699 | 0.811 |
| SNGP | 0.423 | 0.041 | 0.713 | 0.848 |
| Ensemble | 0.356 | 0.042 | 0.731 | 0.825 |
| BALD | | | | |
| MC-Dr. | 0.401 | 0.046 | 0.698 | 0.800 |
| SNGP | 0.417 | 0.039 | 0.712 | 0.842 |
| Ensemble | 0.350 | 0.040 | 0.724 | 0.817 |
| Variation Ratio | | | | |
| MC-Dr. | 0.401 | 0.048 | 0.701 | 0.807 |
| SNGP | 0.413 | 0.040 | 0.718 | 0.835 |
| Ensemble | 0.343 | 0.039 | 0.732 | 0.828 |

Table 5: *Final / AUC* accuracies (↑) of evaluation models on CIFAR10 for different epistemic strategies and Bayesian methods.

|  | Random | Entropy | BALD | VR |
|---|---|---|---|---|
| MC-Dr. | 0.847 / 0.720 | 0.873 / 0.731 | 0.874 / 0.735 | 0.876 / 0.735 |
| Ensemble | 0.848 / 0.720 | 0.868 / 0.725 | 0.874 / 0.734 | 0.873 / 0.736 |
| SNGP | 0.847 / 0.720 | 0.872 / 0.731 | 0.871 / 0.730 | 0.874 / 0.737 |

Table 6: Final epistemic estimates (AUPR near ↑ / AUPR far ↑) of evaluation models on CIFAR10 for different epistemic strategies and Bayesian methods.

|  | MC-Dropout | Ensemble | SNGP |
|---|---|---|---|
| Random | 0.761 / 0.880 | 0.758 / 0.881 | 0.759 / 0.874 |
| Entropy | 0.784 / 0.897 | 0.782 / 0.900 | 0.784 / 0.909 |
| BALD | 0.782 / 0.885 | 0.778 / 0.880 | 0.786 / 0.895 |
| Variation Ratio | 0.786 / 0.898 | 0.786 / 0.901 | 0.788 / 0.895 |

Table 7: Final aleatoric estimates (Brier ↓ / TCE ↓ / ACE ↓) of evaluation models on CIFAR10 for different epistemic strategies and calibration methods.

|  | MC-Dropout | Ensemble | SNGP |
|---|---|---|---|
| Random | 0.225 / 0.014 / 0.022 | 0.226 / 0.014 / 0.022 | 0.226 / 0.014 / 0.022 |
| Entropy | 0.196 / 0.017 / 0.023 | 0.203 / 0.018 / 0.023 | 0.197 / 0.018 / 0.022 |
| BALD | 0.193 / 0.016 / 0.022 | 0.194 / 0.021 / 0.022 | 0.195 / 0.014 / 0.021 |
| Variation Ratio | 0.189 / 0.017 / 0.022 | 0.195 / 0.018 / 0.023 | 0.192 / 0.015 / 0.021 |

SNGP. However, looking at Tab. 5, we hardly notice any changes in evaluation accuracy across different combinations of Bayesian methods and selection strategies. This suggests that incorporating epistemic uncertainty and the quality of its estimates do not impact instance selection in AL. This observation does not match our motivation from H3 and the intuition from previous work with tabular data (Nguyen et al., 2019). We believe the problem lies in the supervised representation learning process of DNNs. In AL, where DNNs rely on a small annotated labeled pool, these representations might not correctly capture the underlying problem. Hence, exploration does not necessarily lead to a better selection. Still, these findings align with recent research in deep AL, showing no consistent benefits of incorporating epistemic uncertainty into AL (Munjal et al., 2022; Zhan et al., 2022). This is particularly prominent in our study by evaluating strategies largely independent of the selection model. For example, an ensemble of DNNs might deliver a better learning curve than a single model regardless of the instance selection. Moreover, our experiments highlight that the quality of epistemic estimates does not affect selection.

***Takeaway: H3 is invalid.*** *Epistemic strategies do not lead to performance improvements over aleatoric strategies in AL. Further, the quality of epistemic uncertainty estimates seems to have negligible effects.*

**H4:** We find an improved Brier score by looking at the aleatoric uncertainty estimates in Tab. 7. However, unlike aleatoric strategies, we do not see an improved calibration (TCE and ACE). This discrepancy might result from the Brier score's broader evaluation, considering both improved generalization and calibration, while the TCE and ACE solely address the calibration quality. Furthermore, since Bayesian models provide a stronger regularization than deterministic models, focusing on inherently noisy instances might not yield the same benefits. This can also be seen in Appendix C with datasets that have a larger number of classes. We can observe this also for the TCE, which appears slightly worse. Thus, we assume that the improvement of the Brier score is insufficient to conclude improved aleatoric uncertainty estimates. Regarding epistemic uncertainty, we notice that epistemic strategies mostly yield models that provide improved epistemic uncertainty estimates, indicated by the improved AUPR values for near and far OOD detection in Tab. 6.

*__Takeaway: H4 is partially valid.__ Epistemic strategies lead to improved epistemic uncertainty estimates. Additionally, they lead to a slightly worse estimation of the aleatoric uncertainty. This may be due to the already strong regularization and good aleatoric uncertainty estimation of Bayesian models.)*

## 5 Conclusion

We demonstrated that improving aleatoric and epistemic estimates does not necessarily improve AL strategies' instance selection, that the incorporation of epistemic uncertainty has negligible effects, and that uncertainty-based strategies enhance predictive uncertainty. Furthermore, our proposed toolbox for deep AL includes several AL strategies and models for uncertainty modeling to facilitate future studies.

As our study suggests that good uncertainty estimates have a small influence on uncertainty-based strategies, we advise future research to focus on a combination with the learned feature representations of a DNN. More recent selection strategies have shown that this has great potential (Ash et al., 2020; 2021). Orthogonal to the topics discussed in this work, we encourage to study the effect of decoupling AL and the feature learning process within DNNs. Especially in the early stages, AL is often ineffective because DNNs cannot efficiently learn representations with small amounts of labeled data. We believe this poor representation is a significant challenge hindering AL strategies' efficacy. Recent advances in self-supervised learning offer a promising avenue for AL strategies to be more effective (Hacohen et al., 2022).

Regarding practical applicability, a more comprehensive evaluation of uncertainty-based AL strategies is required. On the one hand, when comparing selection strategies, it is essential to assess the instance selection independently of the model. This enables a better understanding of selection strategies and facilitates their practicability. In our work, we realized this through an evaluation model. However, this selection cannot be considered fully independent since an evaluation model might have similarities to a selection model. Future research should consider further techniques for an independent evaluation. On the other hand, we recommend that novel strategies exploiting uncertainty include an evaluation of the uncertainty estimates of the selection model via proper scoring rules, calibration errors, or OOD metrics. This facilitates reproducibility and easier deployment of AL strategies by providing a requisite uncertainty estimate quality. Furthermore, practitioners can more easily decide which models suit a particular task.

Lastly, research should also focus on AL's impact on predictive uncertainty estimates after AL (H4). As datasets obtained through AL are used to train models intended for deployment, ensuring that these models provide precise predictive uncertainty estimates is vital. Consequently, AL strategies must deliver both satisfactory performance and accurate uncertainty estimation.

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

## A  Metrics

This appendix introduces the evaluation metrics used throughout the evaluation study. For each metric, we compute two types of results, namely final and area under the curve (AUC). The first result type refers to the metric score evaluated at the end of the AL process, while the second result type refers to the mean as a proxy of the AUC metric scores evaluated during the AL process. We compute these metrics via a test set $\mathcal{E}$ consisting of tuples $(y, \hat{\boldsymbol{p}}) \in \mathcal{E}$, where $y \in \mathcal{Y}$ denotes the true label of an instance and $\hat{\boldsymbol{p}} = (\hat{p}_1, \ldots, \hat{p}_{|\mathcal{Y}|})^{\mathrm{T}}$ the estimated class probabilities.

**Accuracy** is a popular metric for quantifying classification performances and describes the fraction of correctly classified instances:

$$\mathrm{Accuracy}(\mathcal{E}) = \frac{1}{|\mathcal{E}|} \sum_{(y,\hat{\boldsymbol{p}}) \in \mathcal{E}} \delta(y = \hat{y}(\hat{\boldsymbol{p}})), \tag{8}$$

$$\hat{y}(\hat{\boldsymbol{p}}) = \arg\max_{y \in \mathcal{Y}}(\hat{p}_y), \tag{9}$$

where $\delta : \{\mathrm{False}, \mathrm{True}\} \to \{0, 1\}$ is an indicator function returning $\delta(c) = 1$ if a condition $c$ is true and $\delta(c) = 0$ otherwise. We can interpret the function $\hat{y} : [0, 1]^{|\mathcal{Y}|} \to \mathcal{Y}$ as a decision function since it maps the estimated class probabilities to a decision for the class with the highest probability.

**Negative log-likelihood (NLL)** or cross-entropy in the case of classification is a common metric for evaluating predictive distributions. This metric expresses how well the estimated class probabilities match the true class label distribution. Moreover, the NLL is a proper scoring rule since its minimum score corresponds to perfectly estimating the true class label distribution (Ovadia et al., 2019). Mathematically, the NLL can be expressed as:

$$\mathrm{NLL}(\mathcal{E}) = -\frac{1}{|\mathcal{E}|} \sum_{(y,\hat{\boldsymbol{p}}) \in \mathcal{E}} \boldsymbol{y}^{\mathrm{T}} \ln(\hat{\boldsymbol{p}}), \tag{10}$$

where $\boldsymbol{y} \in \{0, 1\}^{|\mathcal{Y}|}$ denotes the one-hot encoded class label $y$ and $\ln(\hat{\boldsymbol{p}})$ the element-wise usage of the natural logarithm on the vector $\hat{\boldsymbol{p}}$.

**Brier score (Brier)** is another proper scoring rule (Ovadia et al., 2019), which measures the mean squared difference between the true labels and the estimated class:

$$\mathrm{Brier}(\mathcal{E}) = \frac{1}{|\mathcal{E}|} \sum_{(y,\hat{\boldsymbol{p}}) \in \mathcal{E}} (\boldsymbol{y} - \hat{\boldsymbol{p}})^{\mathrm{T}}(\boldsymbol{y} - \hat{\boldsymbol{p}}). \tag{11}$$

**Calibration plots** or reliability diagrams graphically present the quality of a model's calibration (Guo et al., 2017). Their idea is to show whether estimated class probabilities are meaningful. For example,

an estimate of $\hat{p}_y = 0.8$ is to indicate that 80 % of all instances with such an estimate actually belong to class $y$. Since the evaluation set $\mathcal{E}$ is finite, while the potential probability estimates in the interval $[0, 1]$ are uncountably infinite many, we need to resort to an approximation. Therefore, we split the probability interval $[0, 1]$ into $G \in \mathbb{N}_{>0}$ adjacent intervals of equal ranges (here, $G = 10$). For the $g$-th ($g \in \{1, \ldots, G\}$) interval, we construct a set $\mathcal{B}_g \subseteq \mathcal{E}$ to contain all tuples where the maximum class probability falls into the $g$-th interval. Such a set is constructed for each interval. As a result, these sets are pairwise disjoint, and their union equals the evaluation set, i.e., $\mathcal{B}_1 \cup \cdots \cup \mathcal{B}_G = \mathcal{E}$. For each bin $\mathcal{B}_g$, we can now compute the accuracy $\text{Accuracy}(\mathcal{B}_g)$ and the confidence

$$\text{Conf}(\mathcal{B}_g) = \frac{1}{|\mathcal{B}_g|} \sum_{(y,\hat{\boldsymbol{p}}) \in \mathcal{B}_g} \max_{y \in \mathcal{Y}} (\hat{p}_y). \tag{12}$$

The final calibration plot is obtained by plotting each tuple $(\text{Conf}(\mathcal{B}_g), \text{Accuracy}(\mathcal{B}_g))$ as a point in a cartesian coordinate system, where the confidences represent the values of the abscissa and the accuracies the values of the ordinate. The connection of these points forms a calibration curve. These top-label calibration plots consider only predictions with maximum class probability and hence neglect all other estimated class probabilities. For this reason, we additionally consider marginal calibration plots, which visualize the mismatch of estimated probabilities for each class. For a class $y \in \mathcal{Y}$, the idea is to define the sets $\mathcal{B}_1^y, \ldots, \mathcal{B}_G^y$ in analogy to the top-label calibration plot with the difference that the separation depends on the probability for class $y$. Accordingly, the sets are also pairwise disjoint, and their union equals the evaluation set, i.e., $\mathcal{B}_1^y \cup \cdots \cup \mathcal{B}_G^y = \mathcal{E}$. Further, the definitions of the confidence and accuracy now depend on class $y$ through:

$$\text{Conf}_y(\mathcal{B}_g^y) = \frac{1}{|\mathcal{B}_g|} \sum_{(y',\hat{\boldsymbol{p}}) \in \mathcal{B}_g^y} \hat{p}_y, \tag{13}$$

$$\text{Accuracy}_y(\mathcal{B}_g^y) = \frac{1}{|\mathcal{B}_g^y|} \sum_{(y',\hat{\boldsymbol{p}}) \in \mathcal{B}_g^y} \delta(y = y'). \tag{14}$$

Each tuple $(\text{Conf}_y(\mathcal{B}_g^y), \text{Accuracy}_y(\mathcal{B}_g^y))$ represents a point in the coordinate system of the marginal calibration plot. This procedure is repeated for each class $y \in \mathcal{Y}$ such that there are $|\mathcal{Y}|$ calibration curves in the end.

**Top-label calibration error (TCE)** summarizes a top-label calibration plot in one score. Concretely, the TCE quantifies the absolute difference between confidences and actual accuracies for each set $\mathcal{B}_g$ and weights this error according to the set's cardinality:

$$\text{TCE}(\mathcal{E}) = \sum_{g=1}^{G} \frac{|\mathcal{B}_g|}{|\mathcal{E}|} |\text{Accuracy}(\mathcal{B}_g) - \text{Conf}(\mathcal{B}_g)|. \tag{15}$$

**Adaptive calibration error (ACE)** is an extension of the TCE and was proposed by Nixon et al. (2019) as a distinct version of the general calibration error that aims to overcome some of the weaknesses of the TCE. First, the ACE introduces an adaptive binning scheme by focusing on regions where most probabilistic predictions lie. Thus, the bin intervals are defined so that they contain an equal number of predictions, improving the estimation of the calibration error. Second, the ACE computes the calibration error across all probabilistic predictions, similar to the marginal calibration plot. Nixon et al. (2019) demonstrated that the ACE behaves more favorably and recommend it over evenly binned calibration errors by experimenting with DNNs on image data.

**Area under the precision-recall curve (AUPR)** refers to a metric assessing the quality of separating in-distribution (ID) from out-of-distribution (OOD) data as proxy for evaluating epistemic uncertainty. Following the literature (Liu et al., 2022), we interpret this separation as a binary classification problem, where OOD instances belong to the positive class and ID instances to the negative class. For making predictions, we compute the entropy of a BNN's predictive distribution, i.e., $H[y|\boldsymbol{x}, \mathcal{L}]$, and set a threshold $\tau \in \mathbb{R}$. An instance with an entropy above this threshold, i.e., $H[y|\boldsymbol{x}, \mathcal{L}] > \tau$, is classified as OOD and as ID otherwise. We evaluate such predictions by computing the precision and recall for multiple (adaptively

determined) thresholds $\tau$. Intuitively, precision is a BNN's ability not to label an instance that is ID as OOD, while recall is the BNN's ability to find all the OOD instances. For each evaluated threshold $\tau$, we obtain a pair of precision and recall values which we plot in a cartesian coordinate system with the precision as the abscissa and the recall as the ordinate. Together, the pairs of precision and recall values form a curve under which we can calculate the area as AUPR.

# B   Methods Improving Uncertainty Estimation

In this appendix, we describe the methods used to improve uncertainty estimation in our experiments. We utilize five uncertainty modeling methods to ensure that our analysis includes varying qualities of aleatoric and epistemic uncertainty estimates. First, we explain the calibration methods label smoothing and mixup. Afterward, we detail the Bayesian methods Monte-Carlo (MC)-Dropout and Ensembles.

## B.1   Calibration Methods

**Label smoothing** was originally introduced by Szegedy et al. (2016) in the context of regularization. It prevents a rapid growth of the largest logit in $f_{\boldsymbol{\omega}}$. As a result, the probabilities obtained from the softmax function do not approach 1 or 0 so fast. Consider the cross-entropy loss function for instance $\boldsymbol{x}_n$ with label $y_n$:

$$L(p(y|\boldsymbol{x}_n, \boldsymbol{\omega}), q(y|\boldsymbol{x}_n)) = -\sum_{k=1}^{K} q(y = k|\boldsymbol{x}_n) \log p(y = k|\boldsymbol{x}_n, \boldsymbol{\omega}), \tag{16}$$

where $q(y|\boldsymbol{x})$ is a distribution assigning probability 1 to the correct label $y_n$ and 0 to the remaining ones. Essentially, label smoothing modifies the distribution $q(y|\boldsymbol{x})$ by employing a convex combination:

$$q(y = k|\boldsymbol{x}) = (1 - \epsilon)\delta(y = k) + \epsilon u(k), \tag{17}$$

where $\epsilon$ is the weight of the convex combination and $u(k)$ is a distribution with uniform probability mass across all classes. Using this mixture distribution instead of the original one-hot encoded one during optimization has regularizing effects and was shown to improve calibration by counteracting overconfidence in DNNs (Müller et al., 2019; Thulasidasan et al., 2019). Additionally, label smoothing enhances robustness against adversarial instances and leads to more discriminative features, supporting transfer learning scenarios.

**Mixup** was introduced by Zhang et al. (2018) as a data augmentation technique for image classification to improve a DNN's generalization and reduce its sensitivity to adversarial instances. During training, it convexly combines random image pairs and corresponding labels as follows:

$$\tilde{\boldsymbol{x}} = \lambda \boldsymbol{x}_i + (1 - \lambda)\boldsymbol{x}_j, \tag{18}$$

$$\tilde{\boldsymbol{y}} = \lambda \boldsymbol{y}_i + (1 - \lambda)\boldsymbol{y}_j. \tag{19}$$

Here, $\boldsymbol{x}_i$ and $\boldsymbol{x_j}$ represent randomly sampled images, while $\boldsymbol{y}_i$ and $\boldsymbol{y}_j$ denote their one-hot-encoded labels. The resulting image-label pair $(\tilde{\boldsymbol{x}}, \tilde{\boldsymbol{y}})$ is then used for training. The coefficient $\lambda \in [0, 1]$ determines the mixing intensity of two images and is sampled from a symmetric Beta distribution $\text{Beta}(\alpha, \alpha)$, where $\alpha \in \mathbb{R}_{>0}$ is a hyperparameter. Generally, smaller values within the range of $\alpha \in [0.1, 0.4]$ lead to the most noticeable performance enhancements. Thulasidasan et al. (2019) found that mixup not only improves classification performance but also enhances the calibration of DNNs. Thus, DNNs trained with mixup exhibit increased robustness against overconfident predictions.

## B.2   Bayesian Methods

**Ensembles** are a collection of several neural networks, called members, and a popular method to obtain a BNN (Lakshminarayanan et al., 2017). Each member is randomly initialized with identical architecture, and member parameters are optimized by maximizing the log posterior, i.e., cross-entropy with weight decay. As

Table 8: *Final / AUC* overconfidence errors (↓) of evaluation models on CIFAR10 for different aleatoric strategies and calibration methods.

|  | - | Label Smoothing | Mixup |
|---|---|---|---|
| Random | 0.055 / 0.115 | 0.023 / 0.048 | 0.009 / 0.021 |
| Least confident | 0.042 / 0.104 | 0.011 / 0.035 | 0.001 / 0.015 |
| Margin | 0.045 / 0.104 | 0.012 / 0.036 | 0.002 / 0.014 |
| Entropy | 0.042 / 0.106 | 0.009 / 0.035 | 0.001 / 0.015 |

each optimized member can be seen as a mode of the posterior distribution, we can interpret each ensemble member as a parameter sample of the true posterior distribution $p(\boldsymbol{\omega}|\mathcal{D})$.

**MC-Dropout**, proposed by Gal & Ghahramani (2016), is a method that employs the regularization technique Dropout (Srivastava et al., 2014) during evaluation to obtain samples from the predictive distribution $p(y|\boldsymbol{x}, \mathcal{D})$. Specifically, dropping out neurons during inference can be interpreted as propagating instances through various neural network versions, which are samples of a distribution approximating the posterior distribution (Gal et al., 2017). Due to MC-Dropout's simplicity and efficiency, it is often used in the literature.

**SNGP** (Liu et al., 2020) is a BNN that uses a last-layer Laplace approximation in combination with spectral normalization (Miyato et al., 2018) and random Fourier features (RFF) (Rahimi & Recht, 2007). With these changes, the BNN is capable of modeling epistemic uncertainty by improving its distance-awareness. Adding spectral normalization to the residual layers of a ResNet leads to a distance-aware feature space. Additionally, the composition of RFF and the last-layer Laplace approximation approximates a Gaussian process with a Gaussian kernel.

## C  Additional Results

This appendix presents the remaining quantitative results regarding two further image benchmark datasets, namely SVHN (Netzer et al., 2011) and CIFAR100 (Krizhevsky, 2009). Furthermore, we present a study investigating the influence of aleatoric strategies on the overconfidence of models on CIFAR10. We refer to App. F for the general experimental setup (e.g., preprocessing of images, usage of OOD datasets).

### C.1  Overconfidence Study

Here, we empirically verify our presumption that aleatoric strategies and their associated selection of inherently noisy instances lead to less overconfident DNNs. For this purpose, we investigate selection models using the *overconfidence error* (OCE) and top-label calibration plots. The OCE is a variant of the TCE that only penalizes overconfident bins and is defined by

$$\text{OCE}(\mathcal{E}) = \sum_{g=1}^{G} \frac{|\mathcal{B}_g|}{|\mathcal{E}|} \left( \text{Conf}(\mathcal{B}_g) \cdot \max \left( \text{Conf}(\mathcal{B}_g) - \text{Accuracy}(\mathcal{B}_g), 0 \right) \right). \tag{20}$$

The remaining setup is identical to the experiments from the main part of the paper.

Table 8 presents the results averaged over ten repetitions. We see that aleatoric strategies consistently lead to a smaller OCE and, thus, to less overconfident DNNs. The top-label calibration plots in Fig. 5 verify this observation. For example, when considering the mixup calibration method, we observe a considerable reduction in high-confident predictions when comparing random selection to aleatoric strategies (cf. number of instances in the last two bins). While the last two bins have a roughly equal number of instance when random selection is used, many more instances have a lower confidence prediction with aleatoric strategies.

### C.2  Additional Results on Real-World Data

Here, we provide further experimental results on SVHN, CIFAR100, and ImageNet100. Similar to Hacohen et al. (2022), we used a subset of ImageNet containing 100 classes. More details can be found in our

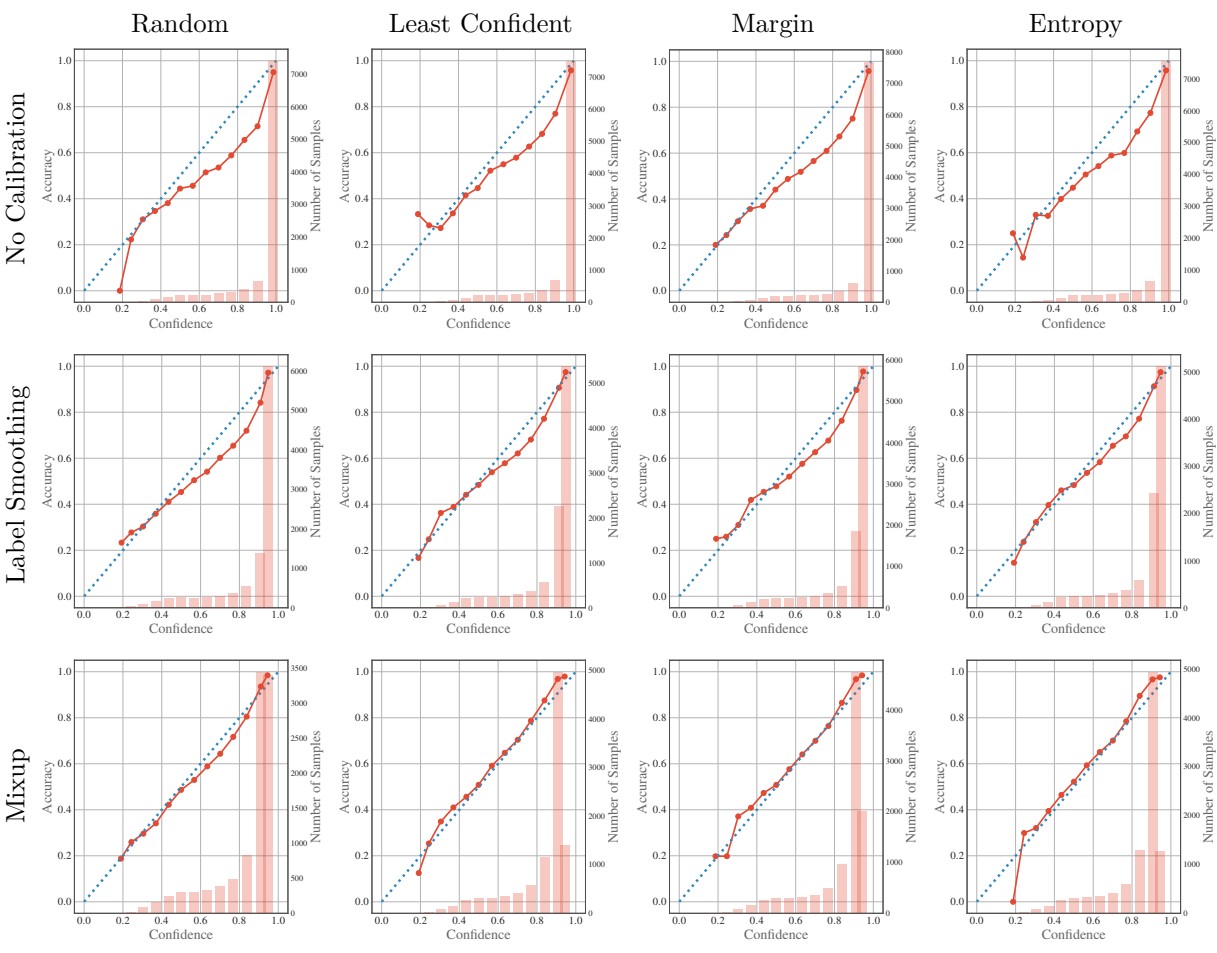

Figure 5: Calibration plots of evaluation models after AL for different selection strategies. We observe that aleatoric strategies lead to less overconfident models, highlighted by the red lines being closer to the (optimal) blue diagonal.

Table 9: Quality of aleatoric uncertainty estimates of selection models on SVHN during AL in form of *AUC*.

| | NLL (↓) | Brier (↓) | TCE (↓) | ACE (↓) |
|---|---|---|---|---|
| | | Random | | |
| - | 0.828 | 0.296 | 0.074 | 0.053 |
| LS | 0.667 | 0.272 | 0.018 | 0.026 |
| Mixup | 0.664 | 0.268 | 0.010 | 0.030 |
| | | Least Confidence | | |
| - | 0.812 | 0.289 | 0.070 | 0.050 |
| LS | 0.654 | 0.266 | 0.016 | 0.027 |
| Mixup | 0.651 | 0.260 | 0.011 | 0.030 |
| | | Margin | | |
| - | 0.775 | 0.275 | 0.065 | 0.047 |
| LS | 0.627 | 0.254 | 0.015 | 0.026 |
| Mixup | 0.623 | 0.248 | 0.011 | 0.029 |
| | | Entropy | | |
| - | 0.833 | 0.296 | 0.072 | 0.051 |
| LS | 0.667 | 0.272 | 0.017 | 0.028 |
| Mixup | 0.671 | 0.269 | 0.012 | 0.030 |

Table 10: *Final / AUC* accuracies (↑) of evaluation models on SVHN for different aleatoric strategies and calibration methods.

| | Random | Least Confident | Margin | Entropy |
|---|---|---|---|---|
| - | 0.914 / 0.807 | 0.926 / 0.812 | 0.929 / 0.821 | 0.925 / 0.808 |
| LS | 0.914 / 0.807 | 0.926 / 0.813 | 0.930 / 0.821 | 0.925 / 0.810 |
| Mixup | 0.913 / 0.805 | 0.927 / 0.810 | 0.928 / 0.818 | 0.924 / 0.804 |

Table 11: *Final* aleatoric estimates (Brier ↓ / TCE ↓ / ACE ↓) of evaluation models on SVHN for different aleatoric strategies and calibration methods.

| | - | Label Smoothing | Mixup |
|---|---|---|---|
| Random | 0.13 / 0.02 / 0.03 | 0.13 / 0.02 / 0.03 | 0.13 / 0.02 / 0.03 |
| Least C. | 0.11 / 0.01 / 0.02 | 0.11 / 0.01 / 0.02 | 0.11 / 0.02 / 0.02 |
| Margin | 0.11 / 0.01 / 0.02 | 0.11 / 0.01 / 0.02 | 0.11 / 0.01 / 0.02 |
| Entropy | 0.12 / 0.01 / 0.02 | 0.12 / 0.01 / 0.02 | 0.12 / 0.02 / 0.03 |

Table 12: Quality of aleatoric and epistemic estimates of Bayesian selection models on SVHN during AL in form of *AUC*.

| | Brier (↓) | ACE (↓) | AUPR (↑) CIFAR10 | AUPR (↑) CIFAR100 |
|---|---|---|---|---|
| | | Random | | |
| MC-Dr. | 0.280 | 0.042 | 0.640 | 0.641 |
| SNGP | 0.375 | 0.056 | 0.693 | 0.697 |
| Ensemble | 0.263 | 0.036 | 0.667 | 0.668 |
| | | Bayesian Entropy | | |
| MC-Dr. | 0.290 | 0.044 | 0.652 | 0.650 |
| SNGP | 0.389 | 0.049 | 0.678 | 0.682 |
| Ensemble | 0.261 | 0.035 | 0.703 | 0.704 |
| | | BALD | | |
| MC-Dr. | 0.285 | 0.041 | 0.648 | 0.648 |
| SNGP | 0.379 | 0.050 | 0.689 | 0.688 |
| Ensemble | 0.251 | 0.033 | 0.700 | 0.698 |
| | | Variation Ratio | | |
| MC-Dr. | 0.269 | 0.039 | 0.664 | 0.661 |
| SNGP | 0.381 | 0.050 | 0.694 | 0.696 |
| Ensemble | 0.246 | 0.033 | 0.714 | 0.712 |

Table 13: *Final / AUC* accuracies (↑) of evaluation models on SVHN for different epistemic strategies and Bayesian methods.

| | Random | Entropy | BALD | VR |
|---|---|---|---|---|
| MC-Dr. | 0.911 / 0.807 | 0.922 / 0.807 | 0.924 / 0.809 | 0.928 / 0.818 |
| Ensemble | 0.911 / 0.807 | 0.922 / 0.802 | 0.927 / 0.812 | 0.929 / 0.815 |
| SNGP | 0.911 / 0.807 | 0.925 / 0.805 | 0.926 / 0.802 | 0.931 / 0.810 |

Table 14: Final epistemic estimates (AUPR near ↑ / AUPR far ↑) of evaluation models on SVHN for different epistemic strategies and Bayesian methods.

| | MC-Dropout | Ensemble | SNGP |
|---|---|---|---|
| Random | 0.765 / 0.762 | 0.765 / 0.762 | 0.765 / 0.762 |
| Entropy | 0.781 / 0.775 | 0.799 / 0.794 | 0.799 / 0.795 |
| BALD | 0.781 / 0.777 | 0.788 / 0.781 | 0.798 / 0.791 |
| Variation Ratio | 0.804 / 0.796 | 0.811 / 0.802 | 0.820 / 0.813 |

Table 15: Final aleatoric estimates (Brier ↓ / TCE ↓ / ACE ↓) of evaluation models on SVHN for different epistemic strategies and Bayesian methods.

| | MC-Dropout | Ensemble | SNGP |
|---|---|---|---|
| Random | 0.138 / 0.044 / 0.027 | 0.138 / 0.044 / 0.027 | 0.138 / 0.044 / 0.027 |
| Entropy | 0.147 / 0.043 / 0.024 | 0.175 / 0.064 / 0.027 | 0.127 / 0.046 / 0.024 |
| BALD | 0.126 / 0.051 / 0.026 | 0.125 / 0.051 / 0.026 | 0.123 / 0.046 / 0.024 |
| Variation Ratio | 0.125 / 0.046 / 0.024 | 0.119 / 0.048 / 0.025 | 0.116 / 0.046 / 0.024 |

implementation. Additionally, we analyze these results in relation to the hypotheses H1 - H4 and briefly summarize our findings. Initially, we evaluate the uncertainty estimates of selection models and examine whether these estimates lead to different evaluation accuracies (H1 and H3) or uncertainty estimates (H2 and H4).

### C.2.1 SVHN

Table 9 demonstrates that calibration methods consistently lead to models that select instances with improved aleatoric estimates compared to the non-calibrated model. Examining Tab. 10, we see that this

Table 16: Quality of aleatoric uncertainty estimates of selection models on CIFAR100 during AL in form of *AUC*.

|  | NLL (↓) | Brier (↓) | TCE (↓) | ACE (↓) |
|---|---|---|---|---|
| Random |||||
| - | 1.685 | 0.533 | 0.030 | 0.034 |
| LS | 1.742 | 0.534 | 0.008 | 0.034 |
| Mixup | 1.656 | 0.523 | 0.001 | 0.033 |
| Least Confidence |||||
| - | 1.612 | 0.523 | 0.030 | 0.036 |
| LS | 1.675 | 0.525 | 0.005 | 0.035 |
| Mixup | 1.597 | 0.516 | 0.001 | 0.034 |
| Margin |||||
| - | 1.611 | 0.521 | 0.029 | 0.035 |
| LS | 1.671 | 0.522 | 0.004 | 0.035 |
| Mixup | 1.599 | 0.514 | 0.000 | 0.033 |
| Entropy |||||
| - | 1.612 | 0.524 | 0.031 | 0.036 |
| LS | 1.671 | 0.524 | 0.005 | 0.035 |
| Mixup | 1.595 | 0.515 | 0.001 | 0.034 |

Table 17: *Final / AUC* accuracies (↑) of evaluation models on CIFAR100 for different aleatoric strategies and calibration methods.

|  | Random | Least Confident | Margin | Entropy |
|---|---|---|---|---|
| - | 0.702 / 0.596 | 0.725 / 0.608 | 0.721 / 0.608 | 0.723 / 0.607 |
| LS | 0.703 / 0.595 | 0.725 / 0.608 | 0.726 / 0.608 | 0.726 / 0.608 |
| Mixup | 0.703 / 0.596 | 0.723 / 0.606 | 0.723 / 0.607 | 0.723 / 0.606 |

Table 18: *Final* aleatoric estimates (NLL ↓ / TCE ↓ / ACE ↓) of evaluation models on CIFAR100 for different aleatoric strategies and calibration methods.

|  | - | Label Smoothing | Mixup |
|---|---|---|---|
| Random | 1.20 / 0.04 / 0.04 | 1.19 / 0.04 / 0.04 | 1.20 / 0.04 / 0.04 |
| Least C. | 1.08 / 0.04 / 0.04 | 1.08 / 0.04 / 0.04 | 1.09 / 0.04 / 0.04 |
| Margin | 1.09 / 0.04 / 0.04 | 1.08 / 0.04 / 0.04 | 1.09 / 0.04 / 0.04 |
| Entropy | 1.08 / 0.04 / 0.04 | 1.08 / 0.04 / 0.04 | 1.09 / 0.04 / 0.04 |

improved aleatoric estimates do not lead to an improvement in evaluation accuracies (H1). Furthermore, Tab. 11 indicates a slight improvement in aleatoric uncertainty estimates by employing aleatoric strategies (H2). Looking at the uncertainty estimation of Bayesian models in Tab. 12, we see different qualities of aleatoric and epistemic estimates, with the ensemble working robust across strategies. Consequently, the instance selection of the ensemble should result in the best evaluation accuracies when employing epistemic strategies. Examining Tab. 13, we do not see this effect. In particular, while the ensemble selects instances by using the best uncertainty estimates, it does not lead to the best evaluation accuracy across the strategies (H3). Considering the aleatoric uncertainty estimates in Tab. 15, we again see inconsistencies between the Brier score and calibration error when employing epistemic strategies (H4). The improved Brier score is potentially due to the better generalization of the model, while the calibration errors remain unchanged. In contrast, when considering the epistemic uncertainty estimates in Tab. 14, we see that epistemic strategies lead to enhanced epistemic uncertainty estimates for each Bayesian model.

### C.2.2 CIFAR100

Table 16 shows that calibration methods lead to models that select instances with different qualities of aleatoric estimates. In contrast to CIFAR-10 and SVHN, calibration methods do not consistently improve all aleatoric uncertainty estimates over the non-calibrated model for CIFAR100. This is potentially due to the more complex classification task with 100 classes. As mixup consistently provides the best aleatoric estimates for each strategy, we investigate if its selection results in the best evaluation accuracy. Examining Tab. 17, we observe that mixup's selection does not lead to an improvement in evaluation accuracy (H1). Considering the aleatoric estimates, Tab. 18 indicates an improved estimation by employing aleatoric strategies (H2). Looking at the uncertainty estimation of Bayesian models in Tab. 19, we see varying qualities of aleatoric and epistemic estimates. While ensemble is providing robust estimates for both aleatoric and epistemic uncertainty, SNGP provides the best epistemic estimates in the far-OOD detection scenario. Therefore, we assume that the instance selection of the ensemble and SNGP should result in better evaluation accuracies compared to MC-Dropout when employing epistemic strategies. Again, examining Tab. 20, we do not see this effect (H3). Considering the aleatoric uncertainty estimates in Tab. 22, we again see inconsistencies between the Brier score and calibration error when employing epistemic strategies (H4). For CIFAR100, the TCE appears to be worse and more sensitive when using epistemic strategies, which might be due to the increased number of classes resulting in challenges of maintaining a proper calibration. When considering the epistemic uncertainty estimates in Tab. 21, epistemic strategies lead to enhanced epistemic uncertainty estimates in far-OOD scenarios (H4). In contrast, the results for epistemic estimates in the near-OOD scenarios are inconsistent. Only SNGP's selection yields improved epistemic uncertainty estimates.

Table 19: Quality of aleatoric and epistemic estimates of Bayesian selection models on CIFAR100 during AL in form of *AUC*.

| | Brier (↓) | ACE (↓) | AUPR (↑) CIFAR10 | AUPR (↑) SVHN |
|---|---|---|---|---|
| Random | | | | |
| MC-Dr. | 0.540 | 0.030 | 0.661 | 0.831 |
| SNGP | 0.627 | 0.032 | 0.654 | 0.867 |
| Ensemble | 0.489 | 0.032 | 0.682 | 0.840 |
| Bayesian Entropy | | | | |
| MC-Dr. | 0.538 | 0.031 | 0.668 | 0.844 |
| SNGP | 0.636 | 0.035 | 0.641 | 0.895 |
| Ensemble | 0.474 | 0.034 | 0.699 | 0.860 |
| BALD | | | | |
| MC-Dr. | 0.534 | 0.030 | 0.666 | 0.839 |
| SNGP | 0.621 | 0.034 | 0.658 | 0.876 |
| Ensemble | 0.474 | 0.033 | 0.691 | 0.846 |
| Variation Ratio | | | | |
| MC-Dr. | 0.533 | 0.030 | 0.668 | 0.841 |
| SNGP | 0.627 | 0.034 | 0.652 | 0.891 |
| Ensemble | 0.472 | 0.033 | 0.696 | 0.858 |

Table 20: *Final / AUC* accuracies (↑) of evaluation models on CIFAR100 for different epistemic strategies and Bayesian methods.

| | Random | Entropy | BALD | VR |
|---|---|---|---|---|
| MC-Dr. | 0.703 / 0.595 | 0.725 / 0.607 | 0.722 / 0.606 | 0.724 / 0.609 |
| Ensemble | 0.701 / 0.595 | 0.728 / 0.606 | 0.723 / 0.607 | 0.725 / 0.607 |
| SNGP | 0.703 / 0.595 | 0.724 / 0.607 | 0.721 / 0.606 | 0.721 / 0.608 |

Table 21: Final epistemic estimates (AUPR near ↑ / AUPR far ↑) of evaluation models on CIFAR100 for different epistemic strategies and Bayesian methods.

| | MC-Dropout | Ensemble | SNGP |
|---|---|---|---|
| Random | 0.706 / 0.898 | 0.706 / 0.899 | 0.706 / 0.878 |
| Entropy | 0.701 / 0.913 | 0.703 / 0.915 | 0.722 / 0.887 |
| BALD | 0.707 / 0.921 | 0.708 / 0.904 | 0.719 / 0.884 |
| Variation Ratio | 0.705 / 0.919 | 0.704 / 0.917 | 0.714 / 0.881 |

Table 22: Final aleatoric estimates (Brier ↓ / TCE ↓ / ACE ↓) of evaluation models on CIFAR100 for different epistemic strategies and Bayesian methods.

| | MC-Dropout | Ensemble | SNGP |
|---|---|---|---|
| Random | 0.434 / 0.124 / 0.035 | 0.433 / 0.124 / 0.035 | 0.414 / 0.053 / 0.043 |
| Entropy | 0.416 / 0.136 / 0.040 | 0.416 / 0.135 / 0.040 | 0.391 / 0.057 / 0.045 |
| BALD | 0.417 / 0.139 / 0.039 | 0.419 / 0.132 / 0.038 | 0.391 / 0.050 / 0.043 |
| Variation Ratio | 0.416 / 0.136 / 0.039 | 0.418 / 0.136 / 0.038 | 0.391 / 0.050 / 0.043 |

Table 23: Quality of aleatoric uncertainty estimates of selection models on ImageNet100 during AL in form of *AUC*.

| | NLL (↓) | Brier (↓) | TCE (↓) | ACE (↓) |
|---|---|---|---|---|
| Random | | | | |
| - | 1.664 | 0.528 | 0.054 | 0.040 |
| LS | 1.710 | 0.532 | 0.046 | 0.035 |
| Mixup | 1.619 | 0.516 | 0.053 | 0.036 |
| Least Confidence | | | | |
| - | 1.606 | 0.527 | 0.055 | 0.042 |
| LS | 1.629 | 0.525 | 0.044 | 0.036 |
| Mixup | 1.587 | 0.527 | 0.053 | 0.037 |
| Margin | | | | |
| - | 1.595 | 0.521 | 0.051 | 0.041 |
| LS | 1.633 | 0.523 | 0.047 | 0.036 |
| Mixup | 1.585 | 0.522 | 0.054 | 0.036 |
| Entropy | | | | |
| - | 1.610 | 0.527 | 0.055 | 0.042 |
| LS | 1.622 | 0.523 | 0.043 | 0.037 |
| Mixup | 1.586 | 0.527 | 0.052 | 0.037 |

Table 24: *Final / AUC* accuracies (↑) of evaluation models on ImageNet100 for different aleatoric strategies and calibration methods.

| | Random | Least Confident | Margin | Entropy |
|---|---|---|---|---|
| - | 0.720 / 0.597 | 0.723 / 0.600 | 0.724 / 0.604 | 0.713 / 0.598 |
| LS | 0.714 / 0.596 | 0.722 / 0.602 | 0.730 / 0.605 | 0.719 / 0.603 |
| Mixup | 0.716 / 0.596 | 0.721 / 0.598 | 0.719 / 0.599 | 0.719 / 0.598 |

Table 25: *Final* aleatoric estimates (NLL ↓ / TCE ↓ / ACE ↓) of evaluation models on ImageNet100 for different aleatoric strategies and calibration methods.

| | - | Label Smoothing | Mixup |
|---|---|---|---|
| Random | 1.11 / 0.03 / 0.04 | 1.12 / 0.03 / 0.04 | 1.11 / 0.03 / 0.04 |
| Least C. | 1.04 / 0.03 / 0.04 | 1.04 / 0.03 / 0.04 | 1.04 / 0.02 / 0.04 |
| Margin | 1.05 / 0.03 / 0.04 | 1.03 / 0.03 / 0.04 | 1.06 / 0.03 / 0.04 |
| Entropy | 1.05 / 0.03 / 0.04 | 1.04 / 0.03 / 0.04 | 1.06 / 0.03 / 0.04 |

### C.2.3 ImageNet100

Table 23 demonstrates that calibration methods during AL yield different qualities of aleatoric estimates. Similar to CIFAR100, calibration methods do not consistently improve all aleatoric uncertainty estimates. As mixup provides the most consistent improvements over the non-calibrated model across metrics, we assume that this improved aleatoric estimates might enhance instance selection. However, Tab. 24 suggests that the instance selection of mixup does not improve the evaluation accuracies (H1). When considering the aleatoric estimates in Tab. 25 of evaluation models, we observe a slight improvement in aleatoric estimates due to an improved NLL (H2). In the context of uncertainty estimation of Bayesian models, Tab. 26 suggests

Table 26: Quality of aleatoric and epistemic estimates of Bayesian selection models on ImageNet100 during AL in form of *AUC*.

|  | Brier (↓) | ACE (↓) | AUPR (↑) CIFAR10 | AUPR (↑) CIFAR100 |
|---|---|---|---|---|
| Random |  |  |  |  |
| MC-Dr. | 0.564 | 0.034 | 0.924 | 0.911 |
| SNGP | 0.597 | 0.035 | 0.910 | 0.907 |
| Ensemble | 0.490 | 0.038 | 0.880 | 0.876 |
| Bayesian Entropy |  |  |  |  |
| MC-Dr. | 0.661 | 0.034 | 0.930 | 0.926 |
| SNGP | 0.638 | 0.038 | 0.924 | 0.922 |
| Ensemble | 0.490 | 0.039 | 0.917 | 0.911 |
| BALD |  |  |  |  |
| MC-Dr. | 0.571 | 0.035 | 0.931 | 0.926 |
| SNGP | 0.598 | 0.037 | 0.919 | 0.919 |
| Ensemble | 0.479 | 0.039 | 0.890 | 0.885 |
| Variation Ratio |  |  |  |  |
| MC-Dr. | 0.612 | 0.036 | 0.937 | 0.932 |
| SNGP | 0.612 | 0.037 | 0.914 | 0.916 |
| Ensemble | 0.481 | 0.039 | 0.910 | 0.903 |

Table 27: *Final / AUC* accuracies (↑) of evaluation models on ImageNet100 for different epistemic strategies and Bayesian methods.

|  | Random | Entropy | BALD | VR |
|---|---|---|---|---|
| MC-Dr. | 0.722 / 0.591 | 0.717 / 0.587 | 0.734 / 0.604 | 0.728 / 0.598 |
| Ensemble | 0.722 / 0.592 | 0.715 / 0.588 | 0.732 / 0.598 | 0.725 / 0.597 |
| SNGP | 0.722 / 0.592 | 0.720 / 0.589 | 0.731 / 0.598 | 0.726 / 0.600 |

Table 28: Final epistemic estimates (AUPR near ↑ / AUPR far ↑) of evaluation models on ImageNet100 for different epistemic strategies and Bayesian methods.

|  | MC-Dropout | Ensemble | SNGP |
|---|---|---|---|
| Random | 0.942 / 0.944 | 0.935 / 0.938 | 0.938 / 0.941 |
| Entropy | 0.942 / 0.943 | 0.923 / 0.928 | 0.935 / 0.934 |
| BALD | 0.957 / 0.958 | 0.956 / 0.955 | 0.940 / 0.942 |
| Variation Ratio | 0.950 / 0.950 | 0.942 / 0.939 | 0.932 / 0.937 |

Table 29: Final aleatoric estimates (Brier ↓ / TCE ↓ / ACE ↓) of evaluation models on ImageNet100 for different epistemic strategies and Bayesian methods.

|  | MC-Dropout | Ensemble | SNGP |
|---|---|---|---|
| Random | 0.403 / 0.137 / 0.040 | 0.402 / 0.135 / 0.040 | 0.406 / 0.134 / 0.041 |
| Entropy | 0.440 / 0.195 / 0.046 | 0.480 / 0.254 / 0.045 | 0.464 / 0.237 / 0.045 |
| BALD | 0.390 / 0.145 / 0.043 | 0.423 / 0.194 / 0.045 | 0.405 / 0.166 / 0.044 |
| Variation Ratio | 0.413 / 0.175 / 0.045 | 0.446 / 0.221 / 0.045 | 0.428 / 0.194 / 0.045 |

that MC-Dropout has better epistemic estimates due to improved out-of-distribution metrics. Despite this, Tab. 27 shows similar accuracies across models indicating that epistemic uncertainty hardly influences the selection (H3). Looking at the aleatoric estimates in Tab. 29, we see that the aleatoric estimates become worse when employing epistemic strategies. This is quite prominent across models and strategies. This behavior is consistent with observations in CIFAR100, likely due to the increased complexity from a large number of classes. Finally, considering the epistemic uncertainty estimates in Tab. 28, we see that BALD and variation ratio mostly improve the epistemic estimates, while this is not the case for entropy sampling. Generally, it seems that having a higher number of classes tends to reduce differences between strategies.

# D   Implementations and Infrastructure

All experiments were conducted using a computer cluster equipped with a combination of NVIDIA Tesla V100 and A100 GPUs. CPU models used in our experiments were Intel Xeon Gold 6252 and AMD EPYC 7662. Each server was equipped with approximately 600 GB of RAM, and the computing infrastructure utilized Ubuntu as an operating system. We implemented all models using Python and leveraged the PyTorch (Paszke et al., 2019) and scikit-learn (Pedregosa et al., 2011) libraries. Additionally, Ray Tune (Liaw et al., 2018) was employed for hyperparameter optimization.

# E   Synthetic Experiments

In our **2D data** example with synthetic data, we sampled 100 instances each from two isotropic Gaussian distributions with mean $\boldsymbol{\mu}_1 = (0,0)^{\mathrm{T}}$ and $\boldsymbol{\mu}_2 = (2,2)^{\mathrm{T}}$. Both have a shared covariance of $\boldsymbol{\Sigma} = 2\boldsymbol{I}$, where $\boldsymbol{I}$ is the identity matrix. Using this setup, we can obtain the true probabilities $p(y|\boldsymbol{x})$ through Bayes' theorem, as we have access to likelihood $p(\boldsymbol{x}|y)$, prior $p(y)$, and marginal likelihood $p(\boldsymbol{x})$. We use the true probabilities as our ground truth aleatoric uncertainty estimates. Considering epistemic uncertainty, obtaining ground truth estimates is not possible, as these depend on the lack of knowledge of a model. We approximate it

Table 30: Dataset overview and settings.

| Dataset | \|Train\| | \|Test\| | \|Val\| | \|Classes\| | OOD |
|---------|-----------|----------|---------|-------------|-----|
| CIFAR10 | 50,000 | 10,000 | 10% | 10 | CIFAR100, SVHN |
| CIFAR100 | 50,000 | 10,000 | 10% | 100 | CIFAR10, SVHN |
| SVHN | 73,257 | 26,032 | 10% | 10 | CIFAR10, CIFAR100 |
| ImageNet100 | 128,545 | 5,000 | 10% | 100 | CIFAR10, CIFAR100 |

using Gaussian Processes because they are often considered the gold standard for uncertainty quantification with low-dimensional datasets (Liu et al., 2022). Consequently, the predictions of a Gaussian process trained on the labeled pool $\mathcal{L}$ are our ground truth estimates for epistemic uncertainty.

# F Data Preparation and Model Training

We follow the ResNet work from He et al. (2016) and apply simple data augmentations for our datasets that include horizontal flips and random cropping with a 4x4 padding.

We perform our experiments on commonly used real-world image classification **datasets**, i.e., CI-FAR10 (Krizhevsky, 2009), CIFAR100 (Krizhevsky, 2009), SVHN (Steet View House Number) (Netzer et al., 2011), and ImageNet (Deng et al., 2009). CIFAR10 consists of 60,000 RGB images, each with a size of 32x32, assigned to one of 10 classes. It comes with a predefined training split of 50,000 and a test split of 10,000 instances. CIFAR100 has the same characteristics as CIFAR10, with the main difference of a bigger class cardinality of 100 classes, resulting in fewer instances per class. SVHN consists of approximately 100,000 Google Street View images of house numbers, each with a size of 32x32, assigned to one of 10 classes. The predefined train split contains 73,257 instances and the test split contains 26,032. Table 30 provides an overview of all real-world dataset settings used in the experiments. ImageNet is a large-scale dataset of over 14 million high-resolution images, each labeled with one of 20,000 possible categories. Unlike CIFAR10 and CIFAR100, ImageNet images vary in size and aspect ratio. In our experiments, we employ a subset of ImageNet-1k following (Hacohen et al., 2022), focusing on 100 classes. It comprises a training split of 128,545 images and a test split with 5,000 images.

In every cycle, we train a DNN for 200 epochs with a batch size of 32. We optimize **hyperparameters** before starting an AL experiment to establish an optimal setting and avoid bad aleatoric and epistemic uncertainty estimates due to training problems of a model. Although this setting is not possible when employing AL in the real world, it allows us to ensure a better estimation of aleatoric and epistemic uncertainty at the end of the AL process. In particular, we noticed that hyperparameters of methods aiming to improve the predictive uncertainty (e.g., the amount of label smoothing) required careful tuning in conjunction with learning rate and weight decay. To this end, we created a separate validation split comprising 10% of the training split (e.g., 5,000 instances for CIFAR10). After that, we randomly sampled a subset of the training split (including instances and labels), with the number of instances corresponding to an AL experiment's total budget. With this setup, we performed a grid search and chose the hyperparameters that optimized the validation NLL. All remaining instances from the training split (without validation split and training dataset) were used to perform the experiments.

- **Standard:** We optimized learning rate and weight decay:
  - learning rate: $\{0.001, 0.01, 0.1\}$,
  - weight decay: $\{0.0005, 0.005, 0.05\}$.

- **Label smoothing:** We optimized learning rate, weight decay, and label smoothing hyperparameter:
  - learning rate: $\{0.001, 0.01, 0.1\}$,
  - weight decay: $\{0.0005, 0.005, 0.05\}$,
  - label smoothing: $\epsilon \in \{0.05, 0.1, 0.15, 0.2\}$.

- **Mixup:** We optimized learning rate, weight decay, and mixup hyperparameter:

- – learning rate: $\{0.001, 0.01, 0.1\}$
- – weight decay: $\{0.0005, 0.005, 0.05\}$
- – mixup: $\alpha \in \{0.1, 0.15, 0.2, 0.25, 0.3, 0.35, 0.4\}$.

- **MC-Dropout:** We optimized learning rate, weight decay, and dropout rate with 100 as the number of dropout samples:

  - – learning rate: $\{0.001, 0.01, 0.1\}$,
  - – weight decay: $\{0.0005, 0.005, 0.05\}$,
  - – dropout rate: $\{0.1, 0.2, 0.3, 0.4, 0.5\}$.

- **Ensemble:** We optimized learning rate and weight decay with 10 as the number of ensemble members:

  - – learning rate: $\{0.001, 0.01, 0.1\}$,
  - – weight decay: $\{0.0005, 0.005, 0.05\}$.

- **SNGP:** We optimized learning rate and weight decay. The remaining hyperparameters such as kernel scale ($= 1$) and mean field factor ($= 20$) were selected by considering the recommended values in Liu et al. (2022).

  - – learning rate: $\{0.001, 0.01, 0.1\}$,
  - – weight decay: $\{0.0005, 0.005, 0.05\}$.

