# OpenReview forum: "The Interplay of Uncertainty Modeling and Deep Active Learning: An Empirical Analysis in Image Classification"
_TMLR — Accepted by TMLR_

### Review · Reviewer_iLhd · 2023-12-13

**Summary Of Contributions:**

The authors present four hypotheses exploring the interplay between prediction uncertainty (aleatoric and epistemic) and Active Learning (AL) performance. The first hypothesis posits that accurately estimating aleatoric uncertainty might hurt AL data selection. The second suggests that aleatoric-focused AL strategies could enhance model uncertainty by mitigating overconfident predictions. The third hypothesis proposes that epistemic uncertainty can boost AL performance. Lastly, the fourth hypothesis argues that strategies centered on epistemic uncertainty could improve the overall predictive uncertainty of the model.

The authors conduct extensive experiments to test these hypotheses, utilizing both synthetic and real-world data. These experiments serve to either corroborate or refute the proposed hypotheses. The topic of this paper is relatively interesting and proposes a novel perspective for Active Learning.

**Audience:**

Yes

**Claims And Evidence:**

Yes

**Requested Changes:**

See Weakness above.

**Strengths And Weaknesses:**

**Strength**

* [contribution] Although there have been many uncertainty-based active learning in previous works, a detailed analysis of the role of different types of uncertainties (i.e, aleatoric, epistemic) is not fully discussed in previous works. The research problem is novel and interesting.

* [presentation] Paper writing is clear and easy to understand.

* [soundness] The key claims are well supported by the evidence from the experiments.

**Weakness**

* [minor] I appreciate the thorough analysis provided in this study. However, I was curious about the decision to exclude ImageNet from the dataset. If this was due to computational resource limitations, perhaps considering a smaller version of ImageNet could be beneficial? This might enhance the robustness of the findings and provide a more comprehensive understanding.

* [minor] The conclusions are empirical rather than theoretical. It is okay if the method is motivated merely on empirical observations, but the lack of theories could decrease the contributions of the work.

---

> ### Author Response · Authors · 2024-03-06
> **Response to Reviewer iLhd**
>
> Dear Reviewer,
>
> We appreciate your time and commitment to reading our paper and providing an insightful review. We aim to address each of your identified weaknesses and requested changes in the following. For better orientation, we highlight your comments in **bold** and mark our changes/additions in the paper in red.
>
> > **I appreciate the thorough analysis provided in this study. However, I was curious about the decision to exclude ImageNet from the dataset. If this was due to computational resource limitations, perhaps considering a smaller version of ImageNet could be beneficial? This might enhance the robustness of the findings and provide a more comprehensive understanding.**
>
>
> We agree with your suggestion that including ImageNet will be a valuable addition to our work. Initially, as you already correctly guessed, ImageNet was omitted due to constraints on computational resources. However, following your advice, we have now incorporated a scaled-down version of ImageNet, akin to the approach outlined in [1]. Furthermore, we've included a dedicated section in the appendix to discuss the results, specifically addressing our hypotheses.
>
> > **The conclusions are empirical rather than theoretical. It is okay if the method is motivated merely on empirical observations, but the lack of theories could decrease the contributions of the work.**
>
> Indeed, our conclusions are grounded in empirical evidence. We have recently discovered a study [2] exploring a similar topic from a theoretical perspective. They seem to draw similar conclusions as we have done. Notably, they suggest active learning can function as a form of ad hoc regularization, aligning with our findings on aleatoric uncertainty. We integrated these insights into relevant sections of our work.
>
> [1] G. Hacohen, A. Dekel, and D. Weinshall, “Active Learning on a Budget: Opposite Strategies Suit High and Low Budgets,” in International Conference on Machine Learning, PMLR, 2022, pp. 8175–8195.
>
> [2] S. Farquhar, Y. Gal, and T. Rainforth, “On Statistical Bias In Active Learning: How and When to Fix It,” in International Conference on Learning Representations, 2021.

---

### Review · Reviewer_6yY6 · 2024-01-26

**Summary Of Contributions:**

- This paper tries to understand how (i) uncertainty estimation impacts on active learning and (ii) how the active learning strategies impact on uncertainty estimation on image classification tasks with deep learning architecture.
- Authors introduce 4 hypotheses on the effects of estimating uncertainty on the performance of AL strategies.
- By the experiments on both synthetic and real-world datasets, the authors empirically support or disclaim the four hypotheses that they introduced in this paper.

**Audience:**

Yes

**Claims And Evidence:**

Yes

**Requested Changes:**

- See weakness parts
- In general, it would be great if the authors clearly say which hypotheses are validated by the experiments and which hypotheses are not validated.
- Otherwise, it is hard to read and follow what is the final conclusion of this work.
- Title should be clarified as "on image data" because the results are only based on image data. Otherwise, this paper can be overly sold by other researchers.
- Code link seems incorrect. I cannot access github.com/anonymous.

**Strengths And Weaknesses:**

**(1) Strength**
- The authors first highlight their four hypotheses clearly in the paper. Then, they support / disclaim their hypotheses using various experiments (quite easy to read and follow).
- Various qualitative experiments are quite intuitive. Especially Figure 1 and 3 are quite clear to me to understand the main ideas of this paper.

**(2) Weakness (and questions)**
**1. Hypothesis 1:**
- So, here, the meaning of hypothesis 1 is that good estimation of aleatoric uncertainty does not improve the instance selection. This means that estimation accuracy of aleatoric uncertainty does not matter on the instance selection performance for aleatoric strategies?
- In Table 2 shows that aleatoric strategies show better results than Random. Could we understand this as improving the instance selection? Because we can treat "Random" as worse aleatoric uncertainty estimation? (Like all the samples have the same aleatoric uncertainty)

**2. Hypothesis 2:**
- Is it only applicable for DNN? How about other models which are not based on DNN?

**3. Hypothesis 3:**
- So, based on the experiments, Hypothesis 3 is incorrect?
- It would be good to discuss why the motivation explained in Section 3 does not match with the result? Which part of the "motivation" does not match with the results?

**4. Hypothesis 4:**
- It seems like the takeaways in Section 4.2.2 for H4 is somewhat different from the hypothesis in Section 3.
- In Section 3, epistemic strategies can improve both epistemic and aleatoric uncertainty estimations. But in Section 4.2.2, only epistemic uncertainty is improved.

---

> ### Author Response · Authors · 2024-03-06
> **Response to Reviewer 6yY6 (Part 1)**
>
> Dear Reviewer,
>
> We thank you for taking the time and effort to read our paper and to create a thorough review. We aim to address all your questions and requested changes in the following. For better orientation, we highlight your comments in **bold** and mark our changes/additions in the paper in red.
>
> > **So, here, the meaning of hypothesis 1 is that good estimation of aleatoric uncertainty does not improve the instance selection. This means that estimation accuracy of aleatoric uncertainty does not matter on the instance selection performance for aleatoric strategies?**
>
> The first hypothesis states that an accurate estimation of aleatoric uncertainty does not improve instance selection and might even deteriorate the selection. It is important to note that it does matter how well the aleatoric uncertainty is estimated. As our experiments show, an really accurate aleatoric uncertainty estimation (e.g., indicated by low TCE or ACE) might lead to a worse selection because the instances are too difficult to learn for the model.
>
> > **In Table 2 shows that aleatoric strategies show better results than Random. Could we understand this as improving the instance selection? Because we can treat "Random" as worse aleatoric uncertainty estimation? (Like all the samples have the same aleatoric uncertainty)**
>
>
> That is an interesting interpretation of random sampling. We do not believe it is correct to interpret it this way because a random selection of instances has no relation to an active selection. In our work (and many others), a fundamental assumption is that uncertainty-based information is beneficial for selecting instances. This assumption implies a certain base quality of probabilistic predictions. In our context, this means probabilistic predictions are required to be better than random guesses (i.e., random sampling), utilizing information from the training data distribution. We, therefore, investigate our hypotheses by improving this base quality. Nevertheless, your question highlighted this critical aspect we did not mention. We incorporated this assumption into the background section to clarify this.
>
> > **Hypothesis 2: Is it only applicable for DNN? How about other models which are not based on DNN?**
>
>
> When considering different models, this behavior might change based on the model's complexity. As H2 states, selecting instances mimics regularization, which is especially helpful for DNNs. We recently found an article that concludes something comparable [1]. They show that this regularization might not be beneficial in the case of low model complexity (linear models). We agree that this is an important aspect and clarified this in the motivation of the second hypothesis.
>
> > **So, based on the experiments, Hypothesis 3 is incorrect?**
>
>
> Yes, our experiments suggested that incorporating epistemic uncertainty has negligible effects and does not improve the instance selection. To better clarify whether a hypothesis is correct, we will add respective statements to the results section, as suggested in the "Requested Changes" part of your review.
>
> > **Hypothesis 3: It would be good to discuss why the motivation explained in Section 3 does not match with the result? Which part of the "motivation" does not match with the results?**
>
>
> We agree that this is not sufficiently elaborated and included a discussion on why the results do not match the initial motivation in the results section for H3. The motivation of H3 states that improving/including epistemic estimates does enhance instance selection due to an explorative component. When considering data in an Euclidean space, this does indeed work [2]. We believe that the problem lies in the supervised representation learning process of DNNs. In AL, where DNNs rely on a small annotated labeled pool, these representations might not correctly capture the underlying problem. Hence, exploration won't necessarily lead to a better selection.
>
> > **It seems like the takeaways in Section 4.2.2 for H4 is somewhat different from the hypothesis in Section 3. In Section 3, epistemic strategies can improve both epistemic and aleatoric uncertainty estimations. But in Section 4.2.2, only epistemic uncertainty is improved.**
>
> That is true; we concluded from our empirical experiments that only epistemic uncertainty is improved, while we do not have enough evidence to conclude that aleatoric uncertainty is improved. This is due to the unchanged calibration metrics. Thus, the hypothesis is only partially valid. As mentioned earlier, we agree it is a good idea to state whether a hypothesis is valid.

---

> > ### Author Response · Authors · 2024-03-06
> > **Response to Reviewer 6yY6 (Part 2)**
> >
> > > **Requested Changes:**
> > > **In general, it would be great if the authors clearly say which hypotheses are validated by the experiments and which hypotheses are not validated.Otherwise, it is hard to read and follow what is the final conclusion of this work.**
> >
> >
> > We agree and updated the takeaways in section 4.2.2 to directly indicate whether a hypothesis is valid. Additionally, we added more explanations into the result section that should help with any confusion regarding inconsistencies in motivations (e.g., why a motivation might not result in the expected outcome as in H3).
> >
> > > **Title should be clarified as "on image data" because the results are only based on image data. Otherwise, this paper can be overly sold by other researchers.**
> > >
> >
> > We agree and updated the title by adding “in Image Classification” to be even more precise with the scope of the paper.
> >
> > > **Code link seems incorrect. I cannot access [github.com/anonymous](http://github.com/anonymous).**
> >
> >
> > Due to the double-blind reviewing process, we use a placeholder for the actual github link. The actual link to our repository will be made publicly available after the submission.
> >
> > [1] S. Farquhar, Y. Gal, and T. Rainforth, “On Statistical Bias In Active Learning: How and When to Fix It,” in International Conference on Learning Representations, 2021.
> >
> > [2] V.-L. Nguyen, S. Destercke, and E. Hüllermeier, “Epistemic uncertainty sampling,” in Discovery Science, Springer, 2019, pp. 72–86.

---

### Review · Reviewer_oy1G · 2024-03-02

**Summary Of Contributions:**

The paper empirically analyzes an important problem in deep active learning: how to select the samples in the active learning process? It introduces both synthetic and real datasets to analyze the impact of both sample selection methods based on aleatoric and epistemic uncertainty. It demonstrates some hypotheses on these impacts.

**Audience:**

Yes

**Broader Impact Concerns:**

No broader concerns

**Claims And Evidence:**

No

**Requested Changes:**

It will be helpful to clarify the ambiguity mentioned above and provide experiments with different settings.

**Strengths And Weaknesses:**

Strengths

- The paper introduces both synthetic data and real datasets to analyze the pheromone happening in deep active learning.
- The hypothesis on the mechanism of DAL are reasonable and sound.

Weaknesses:

-  The description of the paper needs clarification. It is unclear how the terminologies "Optimal Aleatoric/epistemic entropy" in Figures 1 and 3 are defined. The definition of model entropy requires the model output, how is an optimal probability p(y|x) defined and leveraged in Eq (4)? Does it mean that we can assume the model output is the GT probability?

- Figure 1 (e) shows a problem that may happen with the selection based on entropy, but it will not happen if selecting based on the margin sampling. It is misleading to use this illustration to support H1.

- The definition of an AL cycle is vague. How many steps/epochs does the model train on the currently labeled pairs within one cycle? Will the performance and conclusion be different if the initial pool size is larger or if the model after the initial cycle is better?

- The differences between Tables 2 and 7 are marginal. It's hard to draw conclusions from them.

---

> ### Author Response · Authors · 2024-03-06
> **Response to Reviewer oy1G**
>
> Dear Reviewer,
>
> Thank you for your time and effort to read our paper and to create a thorough review. We aim to address all your questions and requested changes in the following. For better orientation, we highlight your comments in **bold** and mark our changes/additions in the paper in red.
>
> > **The description of the paper needs clarification. It is unclear how the terminologies "Optimal Aleatoric/epistemic entropy" in Figures 1 and 3 are defined. The definition of model entropy requires the model output, how is an optimal probability p(y|x) defined and leveraged in Eq (4)?**
> >
>
> You're correct in pointing out that model entropy typically uses the model output. However, to compute the entropy, we only require a probability distribution. Thus, we use the ground truth distributions p(y|x) and p(y|x, L) for the computation instead of the distribution outputted by the model p(y|x, w). For example, in an aleatoric case, as we have access to p(x,y) (the data generating process), we have the ground truth probability p(y|x) and use this to compute the entropy and select instances. To make the definition of the optimal probabilities more visible, we emphasize the connection between the data generating distributions by including Bayes theorem in the main text. This is also where we explain these terms. If we misunderstood the question, we would be very grateful if you could elaborate on the question so we can address it better.
>
> > **Does it mean that we can assume the model output is the GT probability?**
> >
>
> Not exactly. We are sampling based on an optimal probability but the model output will not be optimal. We make this distinction more clear in the manuscript.
>
> > **Figure 1 (e) shows a problem that may happen with the selection based on entropy, but it will not happen if selecting based on the margin sampling. It is misleading to use this illustration to support H1.**
> >
>
> We respectfully disagree with this point. Margin sampling will lead to the same behavior because the illustrated example is a two class problem. Thus, it will have an equivalent behavior to entropy sampling. The highly cited AL survey of Burr Settles [1] also confirms this: “For binary classification, entropy-based sampling reduces to the margin and least confident strategies above; in fact all three are equivalent to querying the instance with a class posterior closest to 0.5.” (Settles, 2009, p. 13)
>
> > **The definition of an AL cycle is vague. How many steps/epochs does the model train on the currently labeled pairs within one cycle?**
> >
>
> We agree and clarified this setup in the appendix under the section “Data Preparation and Model Training” to which we refer in the main text. Additionally, all training details can be found in our implementation, which will be made publicly available after the submission.
>
> > **Will the performance and conclusion be different if the initial pool size is larger or if the model after the initial cycle is better?**
> >
>
> The performance and conclusions could vary with a larger initial pool size, introducing a different set of challenges primarily related to the exploration-exploitation trade-off. While a larger pool may mitigate some issues, exploring this extensively falls beyond the scope of our current work due to the substantial increase in experimental complexity it would entail.
>
> > **The differences between Tables 2 and 7 are marginal. It's hard to draw conclusions from them.**
> >
>
> We respectfully disagree with your interpretation in this case. In deep active learning, small differences are quite common [2, 3], yet the results we presented demonstrate clear correlations with our hypotheses.
>
> [1] Burr Settles. Active Learning Literature Survey. Computer Sciences Technical Report 1648, University of Wisconsin–Madison. 2009.
>
> [2] P. Munjal, N. Hayat, M. Hayat, J. Sourati, and S. Khan, “Towards robust and reproducible active learning using neural networks,” in Proceedings of the IEEE/CVF Conference on Computer Vision and Pattern Recognition, 2022, pp. 223–232.
>
> [3] Y. Ji, D. Kaestner, O. Wirth, and C. Wressnegger, “Randomness is the Root of All Evil: More Reliable Evaluation of Deep Active Learning,” in Proceedings of the IEEE/CVF Winter Conference on Applications of Computer Vision, 2023, pp. 3943–3952.

---

### Decision · Action_Editor_ptB9 · 2024-04-07

**Recommendation:** Accept as is

**Comment:**

The reviewers are unanimous in their positive sentiment for this paper. For the reasons stated above, I believe that it has sufficient evidence to support its claims, and is potentially of quite a broad interest to the community.

**Audience:**

The topic of active learning (and the related area of curriculum learning), have received interested for some time. In the context of deep learning, these areas are still actively researched and far from being "solved". I believe this study will be of interesting to AL researchers, but also potentially interesting for more some of the broader community trying to train DNN in data-constrained settings.

This layout (with the four hypotheses) makes it easy for the reader to follow the research questions being asked, and quickly parse the takeaways, which is a plus.

**Claims And Evidence:**

The paper provides a study linking uncertainty estimation and active learning, in particular the authors study the effects of aleatoric/epistemic estimation quality on sample utility, and the effect of selection strategy on uncertainty estimation itself. The authors state clearly the four hypotheses that they are investigating, and provide empirical evidence to support / otherwise these hypotheses.

Overall, the study provides sufficient evidence to draw useful conclusions, both through controlled (toy) data, and image classification. The scope of the experiments is somewhat narrow, with the real-world data focussing on small(ish) scale image classification; however, this scope limitation is acknowledged upfront. Therefore, I believe that the paper does provide compelling evidence for the specific claims made.

---

> ### Author Response · Authors · 2024-05-03
> **Camera-ready version**
>
> Dear Action Editor,
>
> We've uploaded the camera-ready version of our paper. If there are any further requirements, please do not hesitate to inform us. We thank the anonymous reviewers and you for the effort and the insightful comments, which enhanced our work.
>
> Best regards, Authors